# MixAssist: An Audio-Language Dataset for Co-Creative AI Assistance in Music Mixing

**Michael Clemens**\*& **Ana Marasović**
University of Utah
{michael.clemens, ana.marasovic}@utah.edu

## Abstract

While AI presents significant potential for enhancing music mixing and mastering workflows, current research predominantly emphasizes end-to-end automation or generation, often overlooking the collaborative and instructional dimensions vital for co-creative processes. This gap leaves artists, particularly amateurs seeking to develop expertise, underserved. To bridge this, we introduce **MixAssist**, a novel audio-language dataset capturing the situated, multi-turn dialogue between expert and amateur music producers during collaborative mixing sessions. Comprising 431 audio-grounded conversational turns derived from 7 in-depth sessions involving 12 producers, MixAssist provides a unique resource for training and evaluating audio-language models that can comprehend and respond to the complexities of real-world music production dialogues. Our evaluations, including automated LLM-as-a-judge assessments and human expert comparisons, demonstrate that fine-tuning models such as Qwen-Audio on MixAssist can yield promising results, with Qwen significantly outperforming other tested models in generating helpful, contextually relevant mixing advice. By focusing on co-creative instruction grounded in audio context, MixAssist enables the development of intelligent AI assistants designed to support and augment the creative process in music mixing.

## 1 Introduction

The integration of Artificial Intelligence (AI) into music production offers compelling opportunities, from automating technical tasks to potentially augmenting creative workflows (Deruty et al., 2022; Nicholls et al., 2018). Within this domain, music mixing — the intricate process of blending and processing individual tracks into a cohesive whole — remains a critical stage heavily reliant on both technical skill and artistic judgment (Moffat, 2021). While commercial AI tools provide valuable end-to-end automation for mixing and mastering, simplifying complex processes and reducing costs (Moffat, 2021; Vanka et al., 2023), they often function as "black boxes", abstracting the underlying techniques and potentially offering limited pedagogical value for users seeking to improve their own mixing proficiency (Vanka et al., 2023; Li et al., 2024b).

However, a significant and growing segment of users — amateurs and pro-ams (Leadbeater & Miller, 2004)— desire tools that facilitate *learning through doing*, providing contextual guidance and co-creative support within their actual mixing environment (Vanka et al., 2023). Moreover, studies involving novice creators in music production highlight how AI, while accelerating ideation, can compress crucial preparation stages and introduce new challenges in selecting, validating, and integrating AI outputs (Fu et al., 2025). Research in Human-Computer Interaction (HCI) increasingly emphasizes understanding situated practices and user needs in designing human-AI collaborative tools, particularly in creative domains like music (McCormack et al., 2020; Eric et al., 2024).

---

\*Portions of this work were conducted while Michael Clemens was at New Jersey Institute of Technology.

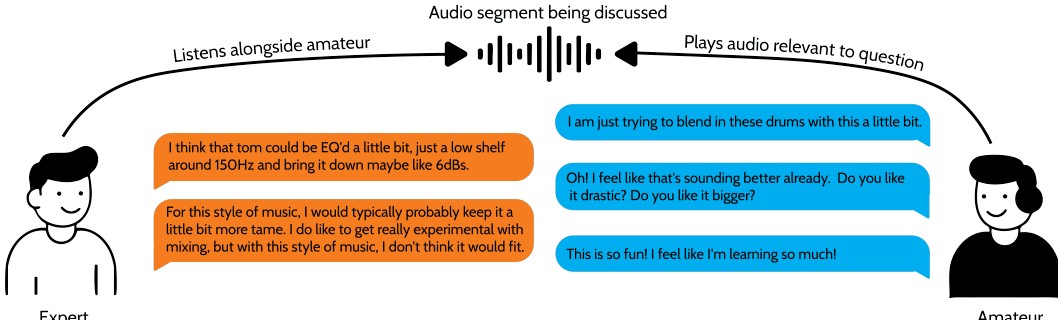

Figure 1: An example of the audio-grounded, multi-turn instructional dialogue captured in the MIXASSIST dataset. An amateur producer plays an audio segment (top) and asks a clarifying question about stylistic choices ("Do you like it drastic?") after receiving specific technical advice (equalization suggestion: "low shelf around 150Hz") from the expert. The expert's response further provides justification based on genre conventions ("For this style of music..."), highlighting the co-creative and pedagogical interaction patterns modeled in the MIXASSIST dataset.

To foster the development of co-creative mixing AI assistants, we introduce **MixAssist**, an audio-language, conversational dataset specifically designed to model the instructional dialogue between expert and amateur music producers. Moving beyond datasets focused on static parameters (De Man & Reiss, 2017), single-turn annotations (captions/tags) (Kim et al., 2019; Drossos et al., 2020; Agostinelli et al., 2023; Law et al., 2009), or general/music-specific question-answering (Gong et al., 2023; Liu et al., 2024; Gardner et al., 2023), MIXASSIST captures the dynamic, turn-by-turn exchange of knowledge grounded in specific audio contexts during live mixing sessions. Figure 1 presents a snippet from one of the sessions, showcasing the interplay between audio context, amateur inquiry, and expert guidance. MIXASSIST contains 431 conversation turns derived from 7 live collaborative mixing sessions involving 12 music producers, featuring temporal alignment between dialogue and the corresponding audio segments being discussed.

Effective music mixing involves both conceptual understanding (the *"why"*, often captured in dialogue) and practical implementation (the *"how"*, reflected in specific settings like equalizations (EQ) levels or compressor thresholds, known as parameters in music production). Our primary focus is on capturing the conversational *"why"* through the audio-grounded dialogues in MIXASSIST. Direct logging of parameters during the data collection sessions was intentionally avoided to preserve the natural workflow and avoid interfering with participants' creative process, as well as to allow participants to use their preferred Digital Audio Workstation (DAW). The nuances related to specific DAW choices and parameter adjustments discussed during these sessions are captured primarily within the dialogue itself (e.g., "I think that guitar should go up about 3dBs").

Separately, we introduce MIXPARAMS, a detailed dataset containing specific parameter settings derived from DAW sessions corresponding to The Mix Evaluation Dataset (De Man & Reiss, 2017), which utilizes the same source songs featured in our MIXASSIST sessions (See Appendix H for more details). While this parallel use of source material suggests potential avenues for future research exploring links between conversational guidance and technical settings, MIXPARAMS serves primarily as a complementary resource. Our primary focus in this work remains the analysis and modeling of the unique conversational dynamics within MIXASSIST. It is this rich, audio-grounded conversational data that provides the foundation for training AI assistants capable of co-creative dialogue.

Leveraging MIXASSIST, we investigate the potential of state-of-the-art audio language models (ALMs) to function as co-creative mixing assistants. We fine-tune three prominent models — Qwen-Audio-Instruct-7B (Chu et al., 2023), LTU (Gong et al., 2023), and MU-LLaMA (Liu et al., 2024) — chosen for their diverse strengths in general audio understanding, audio reasoning, and music-specific processing, respectively. Using a large language model (LLM) ranking generations of these three models, we observe that the fine-tuned Qwen model significantly outperformed the others, achieving the top rank in

50.4% of the evaluations. This finding was validated through human expert assessments, leading to the selection of Qwen for ecologically valid in situ user studies where participants interacted with the model during a mixing task. These subsequent real-time interactions highlighted Qwen's conversational strengths but also revealed significant limitations in its audio analysis capabilities particular for music mixing, demonstrating the necessity of evaluating such assistive creative agents within realistic user workflows (further details in Section 4 and Appendix F).

Our primary contributions through this work include:

- The public release of MixAssist, the first audio-grounded, multi-turn conversational dataset specifically capturing expert-amateur instruction in music mixing.
- Empirical validation of fine-tuned ALMs for conversational mixing assistance, assessed through automated, expert, and in situ user evaluations.

This research aims to foster the development of AI systems that empower artists by enhancing their mixing capabilities with accessible, context-aware guidance, striking a balance between automated support and human creative agency, aligning with calls for more human-centric and collaborative AI tools in creative practices (Tsiros & Palladini, 2020).

## 2 Related Works

**Audio-Language Models (ALMs)** The field of ALMs has evolved rapidly from models learning joint audio-text representations via contrastive learning (e.g., CLAP; Elizalde et al., 2023), which excel at zero-shot classification and retrieval but lack generative capabilities, towards integrating audio perception with the reasoning and generation power of LLMs (Deshmukh et al., 2023). ALMs released in the past few years — including LTU (Gong et al., 2023), Qwen-Audio (Chu et al., 2023), SALMONN (Tang et al., 2023), and Audio Flamingo 2 (Ghosh et al., 2025) — employ stronger LLMs (e.g., LLaMA-7B; Touvron et al., 2023), stronger audio encoders (e.g., AST; Gong et al., 2021), and instruction-finetuning datasets (e.g., OpenAQA-5M; Gong et al., 2023). Specialized models like MU-LLaMA (Liu et al., 2024) and LLARK (Gardner et al., 2023) target music-specific understanding and instruction-following. We select Qwen-Audio-Instruct-7B, LTU, and MU-LLaMA as baselines within our study given their diverse strengths in general audio understanding, audio reasoning, and specialized music processing relevant to the nuances of mixing dialogue.

**AI Music Mixing** AI approaches to music mixing have spanned knowledge-based systems attempting to codify heuristic mixing rules based on genre conventions or perceptual targets (e.g., EQ balancing or stereo placement) (Schaffer & McGee, 1997), data-driven methods learning parameters for effects like EQ or compression (Moffat & Sandler, 2019b; Martinez Ramirez et al., 2021), and newer techniques using differentiable digital signal processing (DDSP) to directly control effects or generate parameters from text (Liu et al., 2023; Chu et al., 2025). While these systems advance automation (Moffat, 2021), they predominantly focus on predicting or applying technical parameters, often lacking conversational features. This contrasts with user studies highlighting a need for assistive, co-creative tools that offer explanation and control, facilitating learning and exploration rather than only end-to-end automation (Vanka et al., 2023). Our work directly addresses this gap by focusing on modeling the explanatory, pedagogical dialogue within the mixing process.

**Text-and-Audio Datasets** Progress in ALMs relies on suitable datasets. Existing resources offer *general audio* classification/tags (e.g., AudioSet; Gemmeke et al., 2017), captions (AudioCaps; Kim et al., 2019), or single-turn QA such as ClothoAQA (Lipping et al., 2022) and OpenAQA-5M (Gong et al., 2023). *Music-specific* data includes captions/tags (MusicCaps; Agostinelli et al., 2023), QA pairs (MusicQA; Liu et al., 2024), or instruction-tuning data (LLARK; Gardner et al., 2023)). *Parameter-focused* datasets like the Mix Evaluation Dataset (De Man & Reiss, 2017) provide technical settings. While these valuable resources capture descriptive labels, static parameters, or single-turn interactions, they generally lack the *dynamic, multi-turn, audio-grounded conversational flow and pedagogical interaction*

inherent in a co-creative, instructional setting such as music mixing. MIXASSIST is the first dataset specifically designed to capture this nuanced dialogue between experts and amateurs during active mixing sessions. This provides a unique resource for training ALMs to understand and participate in the co-creative and instructional aspects of music mixing.

For a detailed discussion of related literature, please see Appendix A.

## 3  MIXASSIST Data Collection

This section details the creation and analysis of the MIXASSIST dataset. We first outline the motivation and design principles guiding the dataset's development (§3.1), then describe the dataset construction process including participant recruitment, session procedure, and data processing steps (§3.2), and finally analyze the resulting data's characteristics.

### 3.1  Motivation and Design Considerations

To ground the development of MIXASSIST, we first surveyed music producers (N=5) with diverse expertise levels about their mixing practices and learning needs (see Appendix B for survey details). The survey confirmed prevalent challenges in music mixing and dissatisfaction with current learning resources among the respondents. There was strong interest in a co-creative AI assistant capable of providing context-aware explanations and guidance, with producers envisioning it as potentially "Very" or "Extremely" useful. Open-ended responses highlighted desires for collaborative development, such as wanting to *"learn from famous mixes"* (Survey Respondent 4 (S4)) or finding *"someone to bounce ideas off of with the goal of fostering development"* (S1). While producers like S4 voiced concerns about adaptability and ethical sourcing ("scraping of techniques"), the clear interest in AI for guided learning motivated our work. Respondent preferences centered on interactive help inside the DAW, utilizing personalized suggestions, examples, and visuals.

These findings motivated us to propose **the task of music-mixing response generation**: Given a conversation history between an amateur and a human/AI expert producer collaboratively mixing music, the amateur producer's latest response, and audio from the most recently played track, the AI assistant must generate a contextually relevant, technically correct, and pedagogically helpful response. To support developing and evaluating ALMs for this task, we identify two key desiderata for a suitable dataset:

- *Instructional Dialog:* Captures the dynamic, multi-turn pedagogical dialogue between experts and amateurs during the mixing process, a dimension missing from prior datasets focused on tags, captions, or single-turn QA.

- *Audio Grounded:* Each conversational turn is associated with a relevant music-only audio segment being discussed.

### 3.2  Dataset Construction & Analysis

We illustrate the process of constructing the MIXASSIST dataset in Figure 2. We first conducted seven hour-long *co-creative mixing sessions.* While these sessions involved expert-amateur pairings, the total number of unique producers was 12 instead of 14, as two individuals participated in multiple sessions swapping roles, allowing us to capture diverse learning dynamics. During the sessions, amateurs mixed multitrack audio recordings sourced from The Mix Evaluation Dataset (De Man & Reiss, 2017), chosen for its permissively licensed material across various genres (see Table 15 for song details). To structure these sessions and elicit realistic pedagogical interactions, we defined distinct participant roles and tasks. Amateurs used their preferred DAW and employed a think-aloud protocol (Van Someren et al., 1994), verbalizing their mixing process and reasoning. Experts acted as mentors, providing guidance primarily upon the amateur's request. This expert-novice pairing and structured interaction protocol aimed to simulate a natural learning environment while ensuring the collected dialogue focused on instructional exchanges relevant to the mixing task. The specific guidelines provided to each participant type are

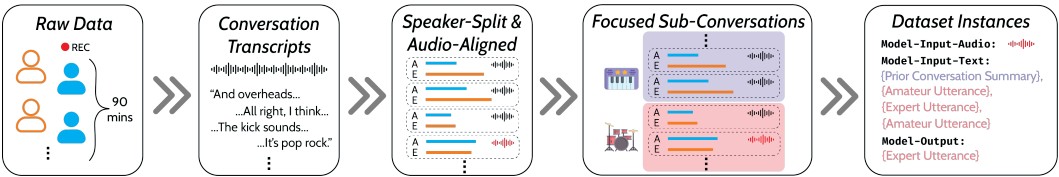

Figure 2: Overview of the MIXASSIST dataset construction pipeline. Raw data from co-creative mixing sessions between expert (E) and amateur (A) producers is first transcribed. These transcripts are then processed to split utterances by speaker and align them with the corresponding audio segments discussed during the session. Aligned turns are grouped into focused sub-conversations based on the mixing topic (e.g., keys, drums, etc). Finally, dataset instances are created for training audio-language models, pairing conversational context (a generated summary and recent dialogue history ending with an amateur utterance) and the relevant audio segment as input, with the subsequent expert utterance as the target output.

detailed in Appendix G.2. The task was for the amateur to complete a mix of the provided song with the expert's guidance. Direct parameter logging was intentionally avoided to maintain ecological validity and allow participants their preferred DAW; nuances related to parameters were captured within the dialogue itself. Full details on participant recruitment, compensation, and other factors are provided in Appendix G. This study was IRB approved.

The next step was to derive dataset instances from the collected ≈7 hours of joint music mixing. Dialog was transcribed using Whisper (Radford et al., 2023) and then manually cleaned by one author, removing filler words while retaining conversational backchannels (e.g., 'uh-huh', 'okay') and using ellipses for natural pauses or fragments. This process included splitting utterances by speaker (amateur/expert) and augmenting dialogue with standardized placeholders (e.g., "Please analyze this audio segment" or "I need more information before I can respond. Please elaborate") where necessary for conversational structure or to account for interaction deviations. Postprocessing also involved removing Personally Identifiable Information (PII) and segmenting relevant music-only audio segments, aligning them with corresponding amateur utterances.

A critical step involved filtering for pedagogical value: Expert responses were manually annotated with a binary 'has_content' tag indicating substantive, actionable guidance. While inherently subjective, this labeling employed reflexive practices (Palaganas & Estacio, 2021; Finlay, 2002) to maintain consistency. This tag was crucial not only for selecting target expert outputs but also for curating the expert utterances within the conversational history provided as input to the models. Specifically, expert turns within the input dialogue history were also filtered to include only those marked 'has_content=True'. This consistent approach was adopted to steer the model's learning towards generating similarly actionable feedback. The final MIXASSIST instances exclusively use expert responses marked 'has_content=True' as the target output, ensuring the dataset focuses on meaningful instructional content.

To structure the data, consecutive utterances were grouped into sub-conversations by mixing focus (e.g., drums, vocals), identified through instrument mentions and dialogue context. These sub-conversations represent coherent instructional episodes. For creating turn-level MIXASSIST instances, summaries providing relevant context from earlier in the session were generated using gpt-4o-mini-2024-07-18 instead of using the entire preceding dialogue. This approach provides focused historical context across topic shifts while maintaining manageable inputs for efficient processing (see Appendix G.3 for prompt details).

Finally, each MIXASSIST instance was constructed. The input consists of the generated prior session summary, the dialogue within the current sub-conversation up to the immediately preceding amateur utterance, and the associated music-only audio segment. The target output is the subsequent expert utterance. This process resulted in the final dataset of 431 expert-authored, audio-grounded instructional responses. Illustrative examples of these final dataset instances can be found in Appendix G.5.

These 431 instances were partitioned into training (241 examples), development (34 examples), and test (156 examples) sets. To rigorously test generalization, the split was performed

using sub-conversations as units, holding out conversations from two complete sessions (the shortest, ID 5 Hard Rock; and second longest, ID 2 Pop Rock) exclusively for the test set. Sub-conversations from the remaining five sessions were distributed across all three splits. This strategy ensures evaluation against unseen producer pairs and promotes testing on diverse genres (see Table 14 and Appendix G.1 for further details).

Detailed statistics on the final dataset splits, linguistic properties (e.g., technical term usage), topic distribution (e.g., drums, overall mix), and temporal dynamics (e.g., evidence of amateur vocabulary acquisition) are presented in Appendix G. Additionally, we also release the complete, unprocessed audio recordings from the sessions, containing participant dialogue and simultaneous DAW playback to support research in areas such as end-to-end conversational speech recognition (Prabhavalkar et al., 2023) or the analysis of fine-grained interaction dynamics. The dataset, including both processed data and raw recordings, will be publicly available.

| MixAssist General Data Statistics | |
| --- | --- |
| Number of mixing sessions | 7 |
| Average # of turns per session | 61.57 |
| Average # of topic switches per conversation | 10.00 |
| Average expert utterance length (# tokens) | 36.57 |
| Average amateur utterance length (# tokens) | 25.39 |
| Average audio segment duration (# seconds) | 19.44 |
| **MixAssist Topic Distribution** | |
| Drums | 40.4% |
| Overall mix | 25.3% |
| Guitars | 15.1% |
| Vocals | 8.6% |
| Bass | 6.5% |
| Keys | 4.2% |

| | Train | Dev | Test |
| --- | --- | --- | --- |
| # Expert Responses | 241 | 34 | 156 |
| # Sub-Conversations | 28 | 11 | 38 |
| Avg. # Turns / Sub-Convers. | 8.61 | 3.09 | 4.11 |

Table 1: Dataset statistics of MixAssist.

## 4 Experiments

This section details our experimental methodology for evaluating ALMs as co-creative mixing assistants using the MixAssist dataset, along with results and discussion. We first introduce the baseline ALMs we fine-tuned (§4.1). Next, we describe our evaluation approach, which includes LLM-as-a-judge, human preference judgments, and qualitative user studies involving real-time interaction (§4.2). We then present the quantitative and qualitative results (§4.3), followed by a discussion interpreting these findings and outlining implications for the future design of AI-assisted music mixing tools ( §4.4).

### 4.1 Baselines

Given the multimodal and instructional nature of the task, appropriate baselines require models capable of processing both audio context and dialogue history. Our primary benchmarks are derived from SOTA ALMs specifically fine-tuned on MixAssist. We selected three distinct ALMs based on their diverse pre-training focus and capabilities relevant to our task: **Qwen-Audio-Instruct-7B** (Chu et al., 2023) for general audio understanding, **LTU** (Gong et al., 2023) for audio reasoning, and **MU-LLaMA** (Liu et al., 2024) for music-specific processing via its MERT encoder (Li et al., 2023). To confirm the necessity of fine-tuning, we also evaluated a zero-shot prompting baseline against our fine-tuned model, the results of which are presented in §4.3.

While prompting offers a potential approach, fine-tuning was chosen to adapt these models more deeply to the specific nuances, terminology, and conversational patterns of pedagogical music mixing dialogue present in the MixAssist dataset, which are less likely to be fully captured by zero-shot or few-shot methods alone. For efficient adaptation, we used LoRA (Hu et al., 2022) to avoid the computational cost associated with updating all model parameters. Models were trained for a single epoch; preliminary experiments with more epochs (e.g., 4) showed signs of overfitting on conversational artifacts (e.g., generating repetitive acknowledgments like 'Mm-hmm'), aligning with recommendations in the original Qwen and LTU fine-tuning documentation. Other hyperparameters followed the default settings provided in the respective model implementations.

| Metric | Qwen | LTU | MU-LLaMA |
|---|---|---|---|
| *Overall Performance* | | | |
| Avg. Rank (Lower is better) | **1.59** | 1.70 | 2.71 |
| Avg. Score (out of 10) | **7.06** | 6.48 | 1.46 |
| Times Ranked #1 | **126 (50.4%)** | 111 (44.4%) | 13 (5.2%) |
| *Rank Distribution* | | | |
| % Rank 1 | 50.4% | 44.4% | 5.2% |
| % Rank 2 | 40.4% | 40.8% | 18.8% |
| % Rank 3 | 9.2% | 14.8% | 76.0% |

Table 2: LLM-as-judge (`o3-mini`) ranking results (250 samples)

## 4.2 Evaluation

Through our experiments, we sought to have both a qualitative and quantitative exploration of the evaluation of these models. Full details of the methodology, survey questions, and results for each stage are available in the appendices referenced below.

**N-Gram Overlap Measurements and LLM-as-a-Judge**  Evaluating generative models for complex, subjective tasks like conversational music mixing advice poses significant challenges regarding scalability and consistency (Li et al., 2024a). Consequently, we employed the LLM-as-a-judge framework (Zheng et al., 2023; Gu et al., 2024) using multiple contemporary models as judges, including `gemma3:4b`, `qwen3:8b`, `llama3.1:8b`, and `o3-mini-2025-01-31`. We used a listwise ranking approach (N=3) for each input prompt, asking the judge to rank the outputs based on technical accuracy, helpfulness, and fluency, forcing a clear preference ordering. This methodology, including prompt design and validation against human raters, represents, to our knowledge, the first use of LLM-as-a-judge in the music mixing context. Detailed methodology and results are presented in Appendix D. Standard automated text generation metrics (e.g., BLEU, ROUGE, BERTScore) were also computed. However, given the known limitations of these metrics for evaluating conversational quality (Sai et al., 2022), we prioritize the LLM-as-a-judge and human evaluation findings.

**Human Evaluation: Generated vs. Expert Responses**  Based on its performance in the multi-model LLM-as-a-judge evaluation (see §4.3), Qwen was selected for the subsequent resource-intensive human preference and real-time studies. Recognizing that automated evaluations like LLM-as-a-judge may not fully capture nuanced human preferences (Zheng et al., 2023), we conducted a direct human preference study to provide a gold-standard comparison between the best fine-tuned model (Qwen) and the original human expert responses from the MIXASSIST dataset. This study involved 10 music producers with varying expertise. Participants compared pairs of responses (ground-truth human expert vs. fine-tuned Qwen output) for 10 prompts randomly selected from the test set, evaluating which response was better given the prompt and audio context, or if both were good/bad. Further details on the methodology and participant instructions are provided in Appendix E.

**Human Evaluation: Real-Time Interaction.**  To assess usability and interaction dynamics in a more ecologically valid setting (Kumar et al., 2024; Subramanian et al., 2023), we conducted a real-time interaction study. 10 music producers interacted with the fine-tuned Qwen agent via a chat interface, using it to get guidance on mixing a track of their choice for approximately 5 minutes. Participants then completed a post-interaction survey assessing conversational naturalness, perceived creative contribution, and novelty of ideas, along with open-ended feedback. Survey questions, results, and analysis are detailed in Appendix F.

### 4.3 Results

Our multi-stage evaluation yielded insights into the capabilities and limitations of fine-tuned ALMs as co-creative mixing assistants.

**Automated Evaluation (LLM-as-a-Judge)** Our automated evaluation leverages a robust LLM-as-a-judge framework to compare our fine-tuned baselines. To ensure the reliability of our findings, we used multiple contemporary models as judges (`gemma3:4b`, `qwen3:8b`, `llama3.1:8b`, and `o3-mini-2025-01-31`). The results consistently showed that the fine-tuned Qwen model outperformed LTU and MU-LLaMA, achieving the best average rank across all judges. This out-

| Preference | Count (N=100) | % |
|---|---|---|
| Generated Better | 40 | 40.0 |
| Human Better | 33 | 33.0 |
| Both Good | 12 | 12.0 |
| Both Bad | 15 | 15.0 |

Table 3: Human Preference: Gen. vs. Exp.

come validated its selection for our human evaluations. We also conducted an experiment comparing our fine-tuned Qwen model against a zero-shot prompted baseline, which demonstrated that fine-tuning on MIXASSIST is necessary for achieving high-quality responses. The detailed methodology and full results tables for these evaluations are presented in Appendix D.

**Human Evaluation (Generated vs. Expert)** We present these results in Table 3. When compared directly against the original human expert responses from the MIXASSIST dataset, human evaluators (N=10) surprisingly showed a slight preference for the responses generated by the fine-tuned Qwen model (preferred 40% of the time) over the original human expert responses (preferred 33% of the time). Qualitative analysis suggests generated outputs may have been favored when providing more detailed explanations (e.g., explaining compression concepts when a user expressed confusion, unlike a human response that changed topic) or more structured, direct answers (e.g., concisely suggesting a separate drum reverb vs. a lengthy human anecdote about reverb workflows). Conversely, human responses often excelled at leveraging implicit context (e.g., correctly interpreting ambiguous user input such as "0.5") or providing quick, natural conversational feedback. Given the limited sample size (N=100 comparisons), these observations are preliminary and warrant further investigation. See Appendix E for detailed breakdown and further discussion including specific examples.

**Human Evaluation (Real-Time Interaction)** Table 13 (Appendix) shows the results of the real-time interaction study (N=10). Participants generally found the Qwen-based agent conversational, with 70% rating the interaction as 'Somewhat natural' or 'Very natural'. 70% also reported the agent suggested novel ideas ('Somewhat' or 'A little') they hadn't considered. However, users perceived their own creative contribution as higher than the agent's (60% rated agent contribution 'Slightly less' or 'Significantly less'). Qualitative feedback highlighted good conversational fluency and adaptability but pointed to limitations in the depth and accuracy of its audio analysis and the creativity of its suggestions. Full survey results and thematic analysis are in Appendix F.

### 4.4 Discussion: Future Design of AI-Assisted Music Mixing

Following the co-creative mixing sessions detailed in §3.2, we conducted semi-structured interviews to further inform the design of co-creative agents. Synthesizing insights from these interviews (detailed in Appendix C.2) and the real-time interaction study (Appendix F), we outline key implications for designing effective AI co-creative partners for music mixing.

**Desired Role: Explainable Teacher and Assistant:** A strong theme emerged favoring AI as an educational tool and technical assistant. Producers expressed desire for agents that "explain industry standards" and demystify processes, like compression (Amateur 3 (A3)). The ideal role was often framed as a guide — "'Hey, this needs a compressor,' but this is how you achieve that."' (Expert 7 (E7)). This educational potential was echoed in the

real-time study, where participants valued the agent's conversational ability (70% found it natural) and its capacity to adapt, with one noting, "I especially liked how when I used non-specific, non-technical terms... the agent matched me" (P1). Beyond explanation, automating tedious tasks like track grouping and initial setup was seen as highly valuable for workflow efficiency (E1, A6, E4). Additionally, users strongly preferred feedback integrated visually within their DAW, wanting the AI to, for example, "show like your computer mouse like dragging, moving it... Like this is what would sound good" (A7), rather than relying solely on text.

**Challenge 1: Audio Understanding:** While conversational interaction was generally positive, our user studies revealed significant limitations in the current model's audio analysis capabilities, a critical factor for a mixing assistant. Several participants noted the agent failing to provide meaningful insights based on the audio provided, with one stating, "It would have been more helpful had it been able to analyze the audio" (P5). This work does not claim to have solved this core research challenge; rather, a key contribution of MIXASSIST is providing a benchmark to diagnose and address this very limitation. To quantify the impact of our dataset, we conducted a manual analysis which confirmed that fine-tuning on MIXASSIST significantly enhances the model's ability to provide substantive, task-relevant, and correct guidance compared to a non-fine-tuned baseline. The detailed quantitative results of this analysis are presented in Appendix F

**Challenge 2: Balancing Guidance with Creativity:** Both studies revealed tension between wanting guidance and preserving creative control. While 70% of real-time study participants felt the agent suggested novel ideas, feedback also indicated a desire for "more precise or little more out-of-the-box" suggestions (P5), suggesting the current guidance might be perceived as too generic or conservative. This resonates with interview concerns about AI leading to homogenized sounds if strictly enforcing "industry standard[s]" (E3) or stifling creativity by adapting too closely to a user's style without flexibility (E4). Designing agents that can offer technically sound advice while also prompting creative exploration and allowing for "happy accidents" (E2) remains a core challenge.

**Challenge 3: Ethical Positioning:** Ethical concerns, particularly around data provenance and attribution, were consistently raised in interviews. There was strong consensus on the need for transparency regarding training data and fair compensation or attribution for creators whose work informs AI recommendations, with suggestions ranging from royalty systems (A2) to source citations (E5) and explicit consent (E1, A7). Developing ethical frameworks is crucial to build trust and ensure responsible AI deployment in creative fields.

**Future Directions:** Based on our findings, future work should prioritize enhancing ALMs' audio grounding to improve their ability to understand and reason about complex audio mixtures beyond basic feature detection when applied to music mixing. Concurrently, developing multimodal interaction paradigms that integrate visual feedback within DAWs, as strongly desired by users, is another potential direction. Future research can leverage the parallel structure between MIXASSIST and the MIXPARAMS dataset (Appendix H) to explore the explicit link between instructional dialogue and concrete mixing parameters, potentially enabling models to translate conversational advice into explicit technical actions. This dialogue-centric approach complements other methods that aim to generate parameters directly from textual descriptions in a zero-shot manner, such as LLM2Fx (Doh et al., 2025). Further exploration is needed in adaptive creativity support for music mixing that balances supportive scaffolding with suggestions that encourage user agency, perhaps via configurable AI intervention. Finally, implementing transparent AI systems, alongside community-accepted ethical guidelines for data usage, attribution, and compensation, remains critical. We acknowledge our findings' generalizability is constrained by the challenge of accessing diverse, high-quality open-source multitracks. The MIXASSIST methodology provides a blueprint for expanding data collection to build more robust assistants capable of learning the nuances of pedagogical and co-creative interaction grounded in audio.

## 5   Conclusion

Music mixing remains a complex craft requiring both technical skill and artistic judgment. While AI offers potential to assist, a gap exists in tools that support co-creative learning and collaboration, particularly for amateur music producers. This work introduced MIXASSIST, a novel audio-language dataset capturing the multi-turn, instructional dialogue between expert and amateur producers during live mixing sessions. Derived from 7 sessions involving 12 producers, MIXASSIST provides 431 audio-grounded conversational turns focused on pedagogical interaction.

Our experiments showed that current ALMs, when fine-tuned on MIXASSIST (with Qwen showing particular promise), can generate contextually relevant conversational mixing advice, sometimes even preferred over original human expert responses in ranked comparisons. However, user studies also highlighted limitations, notably in audio understanding and creative suggestion capabilities, along with a strong user desire for explainable, controllable, visually integrated tools that function as teaching assistants instead of black boxes.

By providing the first dataset focused on situated, multi-turn instructional dialogue in music mixing, MIXASSIST enables future research to address these limitations. It serves as a resource for developing and evaluating the next generation of AI mixing assistants—systems designed not to automate, but to collaboratively empower human creativity and skill development in the art of music production.

### Availability

To support further research, the MIXASSIST dataset, including the processed conversational data, associated audio segments, and the raw session recordings, will be made publicly available upon publication. The MIXASSIST dataset is accessible on Hugging Face, with the audio sessions and transcripts available from Zenodo, and the evaluation code on GitHub.

### Ethics Statement

The development and deployment of MIXASSIST were guided by ethical considerations central to co-creative AI. All 12 participants provided informed consent for the recording and use of their dialogue for research purposes, and all Personally Identifiable Information (PII) was redacted during processing (§G). The dataset is built upon publicly available multitrack recordings from "The Mix Evaluation Dataset," ensuring no copyright infringement (§3.2).

However, we acknowledge that the development of advanced AI assistants carries broader societal risks. While our pedagogical approach aims to augment human skill, such tools could be perceived as a threat to producers who earn income by coaching others. Furthermore, we must be cognizant of risks to creative originality. Research has shown that using LLMs can lead to semantic homogenization at a group level (Anderson et al., 2024). This highlights the challenge that AI assistants must enhance human creativity rather than diminish linguistic diversity or personal style (Chakrabarty et al., 2025).

Despite these valid concerns, we believe the benefits of well-designed, human-centric AI assistants are significant. The goal of MIXASSIST is not to prescribe a single correct path, but to aid the creation of agents that act as collaborative partners to scaffold learning, demystify complex concepts, and empower artists (Gabriel et al., 2024). By focusing on collaborative, human-in-the-loop models, our aim is to develop systems that empower users to develop their unique artistic voice with greater confidence and skill, fostering a co-creative process that values both human ingenuity and AI support.

### Acknowledgements

We wish to express our sincere gratitude to the 12 music producers who generously contributed their time and expertise. Their participation in our co-mixing sessions was essential for the creation of the MIXASSIST dataset. We are also grateful for the foundational work of Brecht De Man and his collaborators on The Mix Evaluation Dataset. Their efforts in making this resource publicly available were essential for our research (De Man & Reiss, 2017).

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

# A   Extended Related Works

This section provides a more detailed overview of literature related to the topics discussed in Section 2.

**Audio-Language Models (ALMs).**   The field of ALMs has evolved rapidly from models learning joint audio-text representations via contrastive learning (e.g., CLAP; Elizalde et al., 2023), which excel at zero-shot classification and retrieval but lack generative capabilities, towards integrating audio perception with the reasoning and generation power of LLMs (Deshmukh et al., 2023). ALMs released in the past few years — including LTU (Gong et al., 2023), Qwen-Audio (Chu et al., 2023), SALMONN (Tang et al., 2023), and Audio Flamingo 2 (Ghosh et al., 2025) — employ stronger LLMs (e.g., LLaMA-7B; Touvron et al., 2023), stronger audio encoders (e.g., AST; Gong et al., 2021), and instruction-finetuning datasets (e.g., OpenAQA-5M; Gong et al., 2023).

Recognizing the unique properties of music, specialized models have also emerged. Microsoft's Muzic project explores AI music understanding and generation, including MusicBERT (Zeng et al., 2021) which uses large-scale pre-training for symbolic music understanding. While much of Muzic focuses on generation or symbolic data (Lv et al., 2023; Lu et al., 2023), it highlights the importance of music-specific pre-training. Bridging acoustic music representation with LLMs, MERT (Li et al., 2023) employs self-supervised learning using both acoustic (RVQ-VAE) and musical (CQT) teachers. Building on this, MU-LLaMA (Liu et al., 2024) connects the MERT encoder to LLaMA for music-specific QA and captioning. Addressing similar goals, LLARK (Gardner et al., 2023) specifically targets multimodal instruction-following for music understanding. The model integrates a pre-trained generative music audio encoder (Jukebox (Dhariwal et al., 2020)) with Llama 2, fine-tuning on a unified dataset created by augmenting open-source music datasets with estimated musical features (key, tempo, chords, beats) and LLM-generated instruction pairs covering music understanding, captioning, and reasoning. Our work selects Qwen-Audio-Instruct-7B, LTU, and MU-LLaMA for fine-tuning. Qwen offers robust general audio capabilities and potential for multi-audio input scenarios, LTU provides a strong foundation in general audio reasoning, and MU-LLaMA offers specialized music understanding via MERT features, potentially capturing finer nuances relevant to mixing.

**AI Music Mixing.**   AI approaches to music mixing have traditionally focused on either mimicking expert knowledge or automating specific processes (Moffat & Sandler, 2019b; Moffat, 2021). Knowledge-based and expert systems attempted to codify mixing rules, while optimization techniques sought ideal parameters based on defined objectives like target loudness or minimal masking (Moffat & Sandler, 2019a). Data-driven methods, leveraging machine learning, learn mixing functions from examples, predicting parameters for effects like EQ, compression, or even performing end-to-end mixing for specific stems like drums (Martinez Ramirez et al., 2021; Steinmetz et al., 2021; Martínez-Ramírez et al., 2022; Koo et al., 2022; ?). Differentiable Digital Signal Processing has enabled novel approaches where neural networks learn to directly control or emulate audio effects (Liu et al., 2023; Ramírez et al., 2021). This allows for style transfer of effects and mixing styles (Vanka et al., 2024; Koo et al., 2023), or generating parameters from natural language, as seen in Text2FX (Chu et al., 2025). Graph neural networks are also being explored, for instance, to model audio effect chains (Lee et al., 2024). However, most existing AI mixing research focuses on predicting parameters or applying effects, often lacking integration with conversational LLMs. User studies like those by (Vanka et al., 2023) highlight that professionals, in particular, desire assistive and co-creative tools that offer control and explanation, rather than black-box automation. They value tools that can handle tedious tasks but also facilitate learning and creative exploration. In addition, much of the data used in developing commercial or academic mixing tools remains proprietary.

**Text-and-Audio Datasets.**   Progress in ALMs and intelligent audio tools heavily relies on suitable datasets. For general audio, resources range from classification/tagging datasets (e.g., AudioSet (Gemmeke et al., 2017), FSD50K (Fonseca et al., 2021)) and captioning collections (e.g., AudioCaps (Kim et al., 2019), Clotho (Drossos et al., 2020)) to single-

turn QA benchmarks (e.g., ClothoAQA (Lipping et al., 2022) ) and large-scale, diverse QA datasets like OpenAQA-5M (Gong et al., 2023). Specific tasks like audio difference explanation also have emerging datasets (e.g., ADIFF (Deshmukh et al., 2025)). For music, datasets such as MusicCaps (Agostinelli et al., 2023), MagnaTagATune (Law et al., 2009), MusicQA (Liu et al., 2024), and LLARK's derived instruction-tuning data (Gardner et al., 2023) provide text annotations (captions, tags, QA pairs). Datasets like the Mix Evaluation Dataset offer valuable parameters alongside perceptual ratings, obtaining fine-grained parameter settings for individual tracks often requires further effort; our own work extends such data by annotating detailed DAW parameters, as described further in Appendix H.

## B  Details of the Survey Gauging Interest in AI-Assisted Music Mixing

An online pilot survey was conducted prior to the main data collection phase to gauge interest in AI-assisted music mixing explanation tools, understand user needs and preferences, and recruit participants for the co-creative mixing sessions that generated the MIXASSIST dataset. The survey was completed by 5 respondents with varying music production experience.

### B.1  Survey Questions

The survey included the following questions:

Q1  Which of the following best describes your level of expertise in music production? (Options: Novice, Hobbyist, Aspiring Professional, Professional, Expert)

Q4  Have you used any software or tools that include Artificial Intelligence (AI) features in your music production? If so, which ones? (Examples may include automatic mixing/mastering suggestions, intelligent EQ adjustments, sample generation, etc.) If you have not used any AI tools in your music production, please leave this section blank. (Options for Other)

Q5  How often do you find yourself struggling with a specific aspect of music mixing (e.g., EQing, compression, etc.)? (Options: Never, Rarely, Occasionally, Often, Always)

Q6  What information sources do you currently use to learn about music mixing techniques? (Select all that apply) (Options: YouTube/Online Tutorials, Books, Music Production Forums, Online Courses, Social Media, Other)

Q7  How satisfied are you with the current methods you use to learn about music mixing? (Scale: Extremely dissatisfied to Extremely satisfied)

Q8  Would you be interested in receiving explanations about music mixing techniques from a creative agent (e.g., an AI co-creative assistant)? (Options: Yes, No, Maybe)

Q9  Imagine a creative agent that can explain music mixing techniques in real-time as you're working on a song. How helpful do you think such an agent would be? (Scale: Not at all useful to Extremely useful)

Q10  Which type of explanations would you find most helpful from a creative agent? (Select all that apply) (Options: Conceptual Explanations, Personalized Suggestions, Comparative Examples, Step-by-Step Walk-Throughs, Visual Representations, Other)

Q11  Do you learn better from technical explanations, audio examples, or visual demonstrations while learning music production concepts?

Q12  Would you prefer the creative agent's explanations to be delivered in text, audio, visual or a combination of all three?

Q15  What features would you like most to see in a co-creative music mixing AI agent? (Select all that apply) (Options: Compatibility, Adaptability, Real-time feedback, Alternative perspectives, Targeted sound sculpting, Skill progression, Reference material integration, In-depth explanations, Other)

Q13 In your own words, describe how explanations from a creative agent could be helpful for your music mixing process. (Open Text)

Q14 Are there any specific concerns you might have about using a creative agent for explanations during music mixing? (Open Text)

Q16 [Recruitment text for follow-up study with email collection field]

### B.2 Survey Results Summary

The initial survey provided valuable insights into producer needs. Key findings from the 5 respondents (S1-S5) who provided detailed answers are summarized below.

**Expertise and Challenges:** The respondents included Professionals (2/5), Aspiring Professionals (2/5), and a Hobbyist (1/5). A majority reported struggling with mixing often (2/5) or always (1/5), with the remaining facing occasional challenges (2/5). This suggests mixing remains a significant hurdle across different experience levels.

**Desired Features and Explanations:** Top desired features among these 5 respondents included 'In-depth explanations' (3/5), 'Adaptability' (2/5), 'Real-time feedback' (3/5), and 'Targeted sound sculpting' (4/5). Preferred explanation types spanned 'Personalized Suggestions' (3/5), 'Comparative Examples' (3/5), 'Step-by-Step Walk-Throughs' (2/5), and 'Visual Representations' (2/5). A strong preference (4/5) emerged for receiving explanations via a combination of text, audio, and visual modalities.

| Area | Summary Finding (N=5) |
| --- | --- |
| Experience | 2 Professional, 2 Aspiring Prof., 1 Hobbyist |
| Challenges | 3/5 struggle Often/Always, 2/5 Occasionally |
| AI Interest | 3/5 Yes; 4/5 interested rated helpfulness Very/Extremely High |
| Top Features | Targeted sculpting (4), In-depth explanations (3), Real-time feedback (3) |
| Explanation Pref. | Personalized (3), Comparative (3); Combination modality (4/5) |

Table 4: Key Survey Findings Summary

Detailed responses, such as those from P1 and P4 mentioned in Section 3.1, highlighted desires for guidance (e.g., learning from famous mixes, having someone to bounce ideas off) and specific concerns regarding adaptability and ethics (e.g., scraping techniques).

## C   Details of the Semi-Structured Interview on Future Design

Following the co-creative mixing sessions described in Section 3.2, semi-structured interviews were conducted with participants to gain deeper insights into their experiences and perspectives on the potential role of AI in music mixing workflows.

### C.1   Semi-Structure Interview Question Guide

- How would you describe the ideal workflow when using a co-creative music mixing agent?
    - Would you like to have it be more of a teacher or a creative partner?
    - Would you use it to learn standard industry techniques or primarily for suggestions or recommendations based on your own musical tastes?
- How do you envision the agent providing feedback or suggestions during the mixing process?
- In what ways could co-creative music-mixing agents help overcome common challenges or limitations in the music-mixing process?

- Do you have any concerns about using a co-creative music mixing agent in terms of originality or artistic expression? If yes, what are these concerns?
- If the co-creative agent produces recommendations that are heavily influenced by specific artists or work in the training data, how do you believe credit and attribution be handled? Should there be a system in place to acknowledge and compensate the original creators? What are your thoughts?

## C.2 Thematic Analysis

Following the co-creative mixing experiment, semi-structured interviews were conducted to understand participant perspectives on integrating AI into the music mixing process. We coded the transcripts using thematic analysis (Smith, 2024) and affinity diagrams (Harboe et al., 2012) to identify recurring patterns and group related concepts regarding producers' desired workflows, interaction preferences, and concerns. One of the authors conducted and transcribed all interviews and constructed the first round of affinity diagrams. This author also extracted additional codes, refined emerging themes, and performed iterations until all responses were captured within an overarching theme. Recognizing that the analysis was conducted by a single author, a reflexive approach was integrated throughout the coding and theme development process to enhance trustworthiness (Palaganas & Estacio, 2021; Finlay, 2002). This involved ongoing critical reflection on potential biases, assumptions, and the researcher's positionality in shaping the interpretation of the data, aligning with recommended practices for rigorous qualitative analysis (Braun & Clarke, 2023).

### C.2.1 Preferred Role of AI: Teacher and Technical Assistant

A dominant theme emerging from the interviews was the preference for AI agents to function primarily as educational tools or technical assistants, rather than fully autonomous creative partners. Participants frequently expressed a desire for AI to explain industry standards, demystify complex processes, and offer guidance grounded in established techniques, particularly when navigating unfamiliar genres or technical challenges. This aligns with research exploring AI as a tool to scaffold learning and augment human capabilities in creative domains (Rigopouli et al., 2025; Long et al., 2021). As E7 stated, the ideal AI would feel more like a guide for technical execution:

> "I picture it more of, yeah, a teacher, right? 'Hey, this is how you do this,' or, 'Hey, this needs a compressor,' but this is how you achieve that." - E7

This sentiment was echoed by others who valued explanations of fundamental concepts like compression (A3) or sought confirmation on genre conventions, like A1 who found it helpful when E1 affirmed decisions about drum prominence in rock music. Producers desired AI to clarify the "why" behind techniques or offer standard approaches, acting as an accessible knowledge base. A7 wanted recommendations focused on technical balance rather than creative direction: "...it'd be in the techniques because what I like is my creativity... So it'd be nice to know like... there's gotta be a balance of high and low." - A7. While some saw potential for co-creation (E4), the emphasis was often on the AI supporting the producer's existing creative vision or providing technical validation (E5), rather than dictating creative choices.

### C.2.2 Ideal Feedback Mechanisms and Interaction

Participants strongly favored feedback mechanisms that were explanatory, visually integrated, and controllable. There was a clear preference for understanding why an AI agent suggests a certain action, aligning with principles of Explainable AI (XAI) which aim to make AI decision-making transparent to users (Bryan-Kinns et al., 2024). A5 explicitly desired this educational aspect: "...having an AI partner who shows me how to do everything, and then when I have questions, being able to ask it, and it shows me, just like you said, with like a visualization and a paragraph..."

Visual feedback integrated directly within the Digital Audio Workstation (DAW) was seen as highly beneficial. Rather than just text, producers wanted the AI agent to highlight

specific frequencies on an EQ, demonstrate parameter changes visually, or point to specific parts of the interface, making the guidance immediately actionable. A7 envisioned this interactive guidance:

> "...I'd want like an example box to pop up right around it and show like your computer mouse like dragging, moving it, dragging, moving it. Like this is what would sound good." - A7

Control over the interaction was also crucial. Many participants preferred the AI to remain passive unless explicitly asked for help (A2, E2), comparing the ideal interaction to a helpful assistant that doesn't intrude unless needed, like Microsoft's less intrusive "Clippy" or an agent that gets *"out of my way until I do something that is kind of out of the norm."* (A5). The ability to turn explanations or suggestions on/off was also mentioned (A2).

### C.2.3  Efficiency, Workflow Automation, and Scaffolding

Producers identified significant potential for AI to enhance workflow efficiency by automating tedious organizational tasks and providing useful starting points. Handling mundane setup processes was seen as a way to preserve creative energy. As E4 noted, *"Mixing is so tedious and meticulous that it's kinda draining, and so to be able to speed up that process with the mundane would be really nice." - E4.* Specific examples included automatically analyzing, classifying, grouping, and color-coding tracks based on instrument type (E1, A6).

Beyond organization, AI was seen as helpful for establishing a baseline mix or providing initial settings, akin to the presets or templates found in tools like iZotope Ozone's mastering assistant (A1). This scaffolding allows producers to bypass some initial setup and move more quickly to creative refinement. A1 articulated this desire for adaptable assistance depending on the task:

> "...if you could sort of tell your AI partner how much help you're wanting with this session or workflow, if you want it to really help like kind of be doing background balancing and stuff at all times so that you can really just focus on your creative vision." - A1

This aligns with the goals of many Creativity Support Tools and AI workflow enhancements designed to handle repetitive tasks, potentially boosting productivity and making the creative process more enjoyable (Shneiderman, 2002; Nakakoji, 2006).

### C.2.4  Concerns: Creativity, Originality, and Adaptability

Despite the enthusiasm for AI assistance, producers consistently raised concerns about its potential impact on creativity and originality. A major worry was that relying too heavily on AI, especially one biased towards "industry standards," could lead to homogenization and stifle the unique artistic voice or the serendipitous "happy accidents" that often fuel creative breakthroughs (E2). E3 worried that an AI strictly enforcing standards could hinder creativity, suggesting that "if you have something that's just going to help you achieve industry standard, it might as well just be a preset." - E3.

There was apprehension about AI learning a user's style too well, potentially limiting exploration into new genres or sounds. E4 expressed this need for balance: *"I do like the idea of adaptive, but the algorithm I think would still need to be somewhat flexible... if it is too adaptive... then I want to do something completely different and that adaptation just doesn't work with it." - E4.* Similarly, E7 explicitly stated, *"I definitely don't want it to learn me... Yeah, I definitely want it to test my assumptions..." - E7.* These concerns mirror broader discussions about AI's role in creative fields – whether it acts as a tool, a collaborator, or a force that could diminish human artistic expression (Miller, 2019). The ideal AI, for many, would need to balance offering guidance with preserving the user's agency and potential for creative deviation (A3).

### C.2.5  Ethics: Attribution, Compensation, and Data Provenance

Ethical considerations surrounding the data used to train AI mixing agents were a significant and recurring theme. Producers emphasized the importance of transparency, attribution, and fair compensation for the original creators whose work, techniques, or styles inform the AI's knowledge and recommendations. This reflects growing concerns across AI development regarding data governance, privacy, and fairness (de Berardinis et al., 2025).

There was a strong consensus that creators whose specific styles or copyrighted material are used, especially if explicitly referenced (e.g., *"make it sound like Flume"* (A1)), should be acknowledged and likely compensated. A2 suggested a royalty-based system:

> *"I feel like people who contribute should get some sort of compensation or royalties off of the program that they made. Similar to album sales, everyone involved gets some sort of piece of it."* - A2

Transparency about data sources was also crucial. E5 valued AI systems that cite their sources, even if imperfectly, to build trust. E2 favored a "learn more" button linking suggestions back to original sources or creators. The need for consent from artists to use their work in training data was also highlighted (E1). While acknowledging the complexity, particularly distinguishing learning from copying (A1), participants felt existing frameworks for copyright and intellectual property should be adapted for AI, ensuring human creators remain central and are treated fairly (A1). As A7 stated, using someone else's hard-earned style without consent felt wrong: *"...to be able to steal, to sound like someone else so much, is, yeah, I definitely don't like it without consent."*

## D  LLM as a Judge

### D.1  Methodology Rationale

Evaluating the quality of outputs from generative language models, particularly for complex and subjective tasks like conversational music mixing advice, poses significant challenges. While human evaluation by domain experts provides high-quality assessments, it often faces limitations in scalability, cost, and consistency (Li et al., 2024a). Consequently, using LLMs to approximate human preferences (LLM-as-a-judge) has emerged as a scalable and increasingly standard evaluation framework (Wang et al., 2025; Zheng et al., 2023; Gu et al., 2024). This approach utilizes the advanced reasoning and contextual understanding capabilities of LLMs to approximate human preferences and judgments in a scalable, cost-effective, and often consistent manner (Li et al., 2024a). Studies have shown strong agreement between LLM judges and human evaluations across various domains and criteria (Wang et al., 2025; Zheng et al., 2023), although limitations such as potential biases and prompt sensitivity must be carefully managed. Common methodologies for LLM-as-a-judge include pointwise scoring (evaluating a single output), pairwise comparison (choosing the better of two outputs), and listwise ranking (ordering multiple outputs) (Wang et al., 2025). Pairwise comparison grounds the judgment by directly comparing two items, often leading to better human agreement and calibration. However, fully ranking multiple items requires multiple pairwise comparisons, increasing cost and potentially suffering from intransitivity or position bias effects if not handled carefully (Wang et al., 2025). Listwise ranking, where the judge orders multiple items simultaneously, offers an alternative that provides maximal context to the judge for comparison (Wang et al., 2025). While potentially more cognitively demanding for the judge and susceptible to its own biases, asking for a direct rank order can be efficient. For evaluating the nuanced quality of conversational music mixing advice—both from our three fine-tuned models (Qwen, LTU, MU-LLaMA) and in comparing our best model to a zero-shot prompted baseline—we needed a method that forces a clear differentiation. We opted to ask the judge (human or LLM) to produce a full rank order for the model outputs generated for each input prompt. This specific form of listwise ranking was chosen because it compels the judge to consider all responses within the same context, forcing a comparative judgment and yielding a complete preference order in a single evaluation step, which aligns with our goal of clearly identifying the best

performing model among the candidates. Therefore, to rigorously evaluate the models, we adopted an LLM-as-a-judge strategy using this ranking approach. Our panel of LLM judges included several strong open-source models: `gemma3`, `qwen3`, `llama3.1`, and `o3-mini`. We utilized this framework for two key evaluations: (1) a three-way ranking of our primary fine-tuned models (Table 5), and (2) a direct comparison between our best fine-tuned model and a zero-shot prompting baseline to validate our approach (Table 6). To the best of our knowledge, this work represents the first application of the LLM-as-a-judge methodology specifically for evaluating conversational agents in the specialized domain of music mixing.

Table 5: LLM-as-a-Judge ranking results for baseline models. We report the percentage of times each model was ranked #1 by various judges and the average rank (lower is better).

| Judge | Qwen | | LTU | | MU-LLaMA | |
|---|---|---|---|---|---|---|
| | % Ranked #1 | Avg. Rank | % Ranked #1 | Avg. Rank | % Ranked #1 | Avg. Rank |
| o3-mini | **50.4%** | **1.59** | 44.4% | 1.70 | 5.2% | 2.71 |
| qwen3:8b | **50.0%** | **1.62** | 40.4% | 1.79 | 9.6% | 2.59 |
| gemma3:4b | **45.2%** | **1.74** | 39.2% | 1.87 | 15.2% | 2.38 |
| llama3.1:8b | **38.0%** | **1.89** | 34.8% | 1.91 | 27.2% | 2.20 |

Table 6: Comparison of Fine-Tuned (FT) vs. Zero-Shot Prompted (Base) Qwen Model.

| Judge | Qwen_FT | | Qwen_Base | |
|---|---|---|---|---|
| | % Preferred | Avg. Rank | % Preferred | Avg. Rank |
| o3-mini | **57.6%** | **1.42** | 42.4% | 1.58 |
| qwen3:8b | **62.4%** | **1.38** | 37.6% | 1.62 |
| gemma:4b | **55.2%** | **1.45** | 44.8% | 1.57 |
| llama3.1:8b | **53.2%** | **1.47** | 46.8% | 1.53 |

## D.2 Human Rater Validation

Before conducting the full evaluation, we performed a validation step to gauge the alignment between the LLM judge and human expert preferences within the music mixing domain. We randomly selected 25 prompts from our test set and generated responses from the fine-tuned versions of Qwen, LTU, and MU-LLaMA for each. Three music producers were then asked to rank the three responses for each prompt from best (Rank 1) to worst (Rank 3), and were provided the same instructions as the LLM. Author 1 of this paper also ranked these responses in accordance with the instructions.

The prompt used for both human raters and the LLM judge is as follows:

Listing 1: LLM-as-a-Judge Prompt

```
You are an impartial judge evaluating how closely each response aligns
    with the way an expert producer would assist an amateur during a
    co-creative music mixing session.

Your PRIMARY goal is to rank three given responses based on the
    following evaluation criteria:

CRITICAL EVALUATION CRITERIA (in order of importance):
1. TECHNICAL ACCURACY - Information should be accurate within a music
    production context
2. HELPFULNESS - Addresses the issue at hand with practical, actionable
    advice or suitable explanation given the  u s e r s  behavior
3. CONVERSATION FLUENCY - Response is natural, concise, and decisive

EXAMPLES:

This is an example of an explanation with GOOD technical accuracy: "By
    pulling down the levels in the 400 to 600 Hz range and adding some
```

air around 1.3k to 2k, you're creating more room for the guitar to breathe and stand out in the mix."

Using the standard ranges of frequencies for different instruments while suggesting an EQ is helpful to amateurs in understanding both the technical implementation as well as the reasoning behind why these techniques are useful. This explanation provides the correct ranges technically as well as providing a reasoning for what adding these elements will do for the mix.

This is an example of an explanation with BAD technical accuracy: "I don't understand what you mean by 'pan the crash and ride'."

Panning is a technique to help increase the stereo image of the mix by moving different instruments around the placement of the stereo field. This explanation does not understand this concept of panning, which is a foundation in mixing. For this reason, this explanation is bad in terms of technical accuracy.

This is an example of an explanation with GOOD helpfulness: "To adjust compression, first, locate the compressor on your DAW (Digital Audio Workstation). It usually has a    Compressor    or    Clipping icon. You can drag the    Wet    control to the left or right to make the compression more or less pronounced. The wet control determines how much of the signal is being compressed."

This explanation describes to the amateur music producer exactly what to do without questioning their creative decisions and provides a greater depth of detail for what a compressor does such that the amateur can then use this knowledge within their exploration of the sonic space.

This is another example of GOOD helpfulness: "I need more information before I can respond. Please elaborate."

In some cases, this can be viewed as helpful such that the expert is asking clarifying questions as opposed to assuming what the amateur meant and providing a reason for something that may not be correct.

This is an example of an explanation with BAD helpfulness: "You can't just say,  I  want a really low reverb,    because you can't just turn it off."

This is another example of BAD helpfulness: "It sounds like you are trying to make the drum loop tighter and more focused, while also reducing its energy or intensity. This could be achieved by adjusting the tempo, adding effects such as compression or EQ, or using a different drum sample with a different sound."

While this explanation may appear helpful on the surface, as opposed to asking a follow-up question before providing an answer, this explanation assumes what the user means instead of providing a clarifying question to resolve the discrepancy before continuing.

This kind of explanation belittles the user and undermines their learning potential throughout this process of mixing. This explanation does not help the user understand what may be wrong with their explanation of the reverb and instead does not encourage learning.

This is an example of an explanation with GOOD naturalness: "You're gonna need a little bit of compression on that."

Slang terms such as gonna, gotta, sick, fire, or many other terms help promote a sense of naturalness within the conversation overall.

```
      These terms and other music production jargon may provide a sense of
      ease and naturalness for the amateur who is receiving instructions.
      This wording helps provide a sense of naturalness throughout the
      conversation flow.

This is an example of an explanation with BAD naturalness: "I
      understand, thank you for letting me know. Is there anything else I
      can help with? If not, have a great day and let me know if you need
      any assistance in the future."

This explanation halts the communication between the expert and the
      amateur entirely and may also deter the amateur from asking
      follow-up questions due to the cold way in which the explanation is
      stated.

This is an example of an explanation with GOOD conciseness: "Yes, a
      compressor could help in controlling the dynamic range and making
      sure the volume levels are even across different parts of the song."

This explanation provides accurate information without being too verbose
      and encourages follow-up questions from the amateur.  Specifics such
      as a compressor and what it does are addressed well without being
      overly wordy.

This is an example of an explanation with BAD conciseness: "Yes, I think
      they could be a bit quieter to better balance with the guitar and
      vocals in the mix. It might also help to adjust the levels of each
      instrument to create more clarity and separation between them.
      Additionally, experimenting with different drum samples or effects
      can add depth and complexity to the overall sound."

This explanation wanders around the main argument and keeps adding
      sentences after initially providing a suggestion at the beginning.
      The addition of these sentences might distract or confuse the
      amateur with too many additions to the central concept.

This is an example of an explanation with GOOD decisiveness: "Yeah, so
      we would put the high pass on guitar three then."

This explanation is succinct and provides a clear and actionable path
      forward for the amateur when implementing this suggestion.

This is an example of an explanation with BAD decisiveness: "That's a
      good question. I think it's more of a matter of what you can do with
      the tube screamer."

This explanation does not provide the amateur with a sense of certainty
      for why something was selected and instead provides a generic
      response overall and turns the onus back toward the amateur for what
      they can do with the effect.

You MUST rank these explanations, even if the difference is slight.

Output ONLY in JSON format: {'rank_1': {A, B, or C}, 'rank_2': {A, B, or
      C}; 'rank_3': {A, B, or C}}; with no explanation.

==================
```

Following this instruction block, the original user prompt and the three model responses (labeled A, B, C in randomized order) were provided to the raters/judge.

The aggregated results from the four human raters are summarized in Table 7. Qwen was clearly preferred, receiving the most Rank 1 votes, followed by LTU, and then MU-LLaMA.

| Model | Rank 1 Count | Rank 2 Count | Rank 3 Count |
|---|---|---|---|
| Qwen | 52 | 30 | 18 |
| LTU | 29 | 39 | 32 |
| MU-LLaMA | 19 | 31 | 50 |

Table 7: Aggregated Human Producer Rankings (25 Samples)

*Note: Counts aggregated across 4 human raters for 25 prompts (100 total rankings per rank).*

### D.3 LLM Judge Validation

We then used the `o3-mini` LLM judge to evaluate the same 25 sets of responses using the identical prompt structure shown above, requesting a 1st, 2nd, and 3rd place ranking in JSON format. As established in prior work, careful prompt design is crucial for effective LLM-as-a-judge evaluation, as performance can be highly sensitive to the phrasing, structure, clarity of criteria, and included examples (Zhuo et al., 2024; Gu et al., 2024; Wei et al., 2024). The prompt used in this study was iteratively refined to clearly define the evaluation criteria—prioritizing technical accuracy, helpfulness, and conversational fluency for the music mixing co-creative context—and provide illustrative positive and negative examples, aiming to mitigate ambiguity and improve judge consistency (Gu et al., 2024; Li et al., 2024a).

The LLM judge's rankings for these 25 samples largely mirrored the human preferences, as shown in Table 8. Qwen was ranked first most often, further validating the LLM judge's capability to reflect expert preferences in this domain for our task using the refined prompt.

| Model | Rank 1 Count | Rank 2 Count | Rank 3 Count |
|---|---|---|---|
| Qwen | 15 | 7 | 3 |
| LTU | 9 | 13 | 3 |
| MU-LLaMA | 1 | 5 | 19 |

Table 8: LLM Judge (`o3-mini`) Rankings (25 Samples)

### D.4 Full LLM Judge Evaluation Results

Confident in the LLM judge's alignment for this task, we proceeded with a full evaluation across 250 test samples derived from the MIXASSIST dataset interactions. The `o3-mini` judge produced a 1st, 2nd, and 3rd place ranking for the three models (Qwen, LTU, MU-LLaMA) for every sample prompt. The aggregated performance metrics derived from these rankings are presented in Table 2.

The results indicate that the fine-tuned Qwen model consistently outperformed the other models, achieving the highest number of Rank 1 placements and the best average rank. LTU performed competitively, securing the second position, while MU-LLaMA significantly lagged behind, being ranked last in 76.0% of the samples.

### D.5 Automated Text Generation Metrics

While human-centric evaluations (LLM-as-a-judge, user studies) were prioritized for assessing the quality of conversational mixing advice, we also computed standard automated text generation metrics comparing the fine-tuned models' outputs against the ground-truth human expert responses in the test set. These metrics include BLEU (Papineni et al., 2002), METEOR (Banerjee & Lavie, 2005), ROUGE-L (Lin, 2004), and BERTScore (Zhang et al., 2019). The results are presented in Table 9. It is worth noting that these metrics, particularly

those based on n-gram overlap, often show limited correlation with human judgments of quality for dialogue and instructional tasks.

| Model | BLEU | BLEU-1 | BLEU-2 | BLEU-3 | BLEU-4 | METEOR | ROUGE-L | BERT-S |
|---|---|---|---|---|---|---|---|---|
| Qwen | **0.0631** | **0.1274** | **0.0556** | **0.0363** | **0.0333** | **0.1504** | **0.1071** | 0.8482 |
| LTU | 0.0240 | 0.0820 | 0.0120 | 0.0021 | 0.0000 | 0.1015 | 0.0895 | 0.8349 |
| MU-LLaMA | 0.0274 | 0.0911 | 0.0157 | 0.0022 | 0.0004 | 0.0932 | 0.0830 | **0.8503** |

Table 9: Comparison of fine-tuned models using automated text generation metrics against human references.

### D.6 Topic-Specific Performance

We further analyzed the LLM judge's preferences based on the primary topic of the user's prompt (e.g., focusing on drums, vocals, overall mix). Table 10 shows the percentage of times each model was ranked #1 for different topics, indicating potential strengths or weaknesses of the models concerning specific musical elements.

| Topic | # Samples | Qwen (% Best) | LTU (% Best) | MU-LLaMA (% Best) |
|---|---|---|---|---|
| Overall Mix | 53 (21.2%) | **66.0%** | 28.3% | 5.7% |
| Drums | 101 (40.4%) | 39.6% | **54.5%** | 5.9% |
| Bass | 19 (7.6%) | **57.9%** | 36.8% | 5.3% |
| Guitars | 45 (18.0%) | **48.9%** | 44.4% | 6.7% |
| Vocals | 28 (11.2%) | **57.1%** | 42.9% | 0.0% |
| Keys | 4 (1.6%) | 50.0% | 50.0% | 0.0% |

Table 10: LLM Judge (`o3-mini`) Rank #1 Percentage by Topic (250 Samples)

While Qwen performed best across most topics, including the 'Overall Mix' category, LTU showed a notable strength in prompts specifically concerning drums. This suggests LTU might have captured nuances related to drum mixing particularly well during fine-tuning. However, Qwen's strong performance in general mix advice and other specific instruments solidified its position as the top-performing model overall for music mixing tasks.

Based on both human expert validation and LLM-as-a-judge evaluation using ranking, Qwen demonstrated a higher performance in generating relevant, accurate, and helpful conversational mixing advice compared to LTU and MU-LLaMA when fine-tuned on the MIXASSIST dataset. Therefore, Qwen was selected as the base model for subsequent experiments and system development in our work.

## E  Human Evaluation Study Details

### E.1  Methodology

To complement the automated LLM-as-a-judge evaluation, we conducted a human evaluation study focusing on the perceived quality of the best-performing model's (Qwen) generated responses compared directly against the ground-truth human expert responses from the MIXASSIST dataset. This comparative evaluation was performed to help understand user preferences and the practical utility of the AI assistant's responses in a music mixing context.

We randomly selected 100 prompts from the test set. For each prompt, we paired the ground-truth human expert response with the response generated by the fine-tuned Qwen model. These pairs were presented to participants in a randomized order, with the human and AI responses randomly assigned to 'Response A' or 'Response B' to mitigate order bias.

A total of 10 music producers were recruited for the study. Participants were compensated with a \$5 Amazon Gift Card for their time. Each participant completed 10 comparison rounds. In each round, they were presented with the conversational prompt, the audio segment associated with the query, and the two responses (A and B).

Participants were asked to evaluate the responses based on which they believed to be the best answer given the prompt and context, selecting one of four options for each pair: "Response A is better," "Response B is better," "Both are good," or "Both are bad."

### E.2 Discussion of Preference Results

The aggregate results across 100 comparisons (10 participants evaluating 10 utterances each) are shown in Table 3. The generated Qwen responses were preferred 40% of the time, while the original human expert responses were preferred 33% of the time. In 12% of cases, participants found both responses good, and in 15%, they found both bad.

| Expertise | N | Generated | Human | Both Good | Both Bad |
|---|---|---|---|---|---|
| Amateur | 1 | 3 | **4** | 2 | 1 |
| Intermediate | 7 | **29** | 24 | 4 | 13 |
| Expert | 2 | **8** | 5 | 6 | 1 |

Table 11: Human Preference by Expertise (Counts)

The aggregate results presented in Table 3 indicate a slight preference among human evaluators for the generated responses from the fine-tuned Qwen model (40.0%) compared to the original human expert responses (33.0%). This finding, while based on a limited sample of 100 comparisons, warrants a closer examination as to why generated responses were rated slightly higher.

A qualitative review of the individual comparison instances where generated responses were preferred suggests potential reasons for this trend:

- **Greater Detail and Explanation:** In some cases, the generated response provided more comprehensive background or step-by-step guidance. For instance, when one user expressed general confusion about compression (Item ID 41), the generated response explained the concept (*"Compression is a technique used... It makes quiet parts louder..."*) and gave starting ratio suggestions, whereas the human response pivoted to specific EQ advice for vocals. Similarly, in Item ID 21, the generated response explained the rationale for general drum compression, while the human response abruptly changed topic to vocals.

- **Structured or Direct Responses:** Generated responses sometimes offered more structured or directly relevant answers. In Item ID 45, the generated response recapped previous steps (*"The first thing we did was adjust the panning..."*) before asking for clarification, perceived as more helpful than the human's slightly vague instruction. In Item ID 74, faced with a question about separate drum reverb, the generated response was a direct *"I think we could do a separate drum reverb,"* while the human expert provided a lengthy, multi-faceted anecdote about personal reverb techniques.

- **Consistent Focus:** Generated responses generally addressed the immediate prompt directly, potentially being perceived as more focused than human responses that might anticipate future steps or change topic.

Conversely, the human expert responses were often preferred when they demonstrated strengths potentially harder for the current models to replicate:

- **Leveraging Implicit Context:** Humans could often interpret ambiguous or under-specified user input. In Item ID 47, the user asked *"What are you thinking, 0.5?"*, which the human expert implicitly understood and confirmed ("Yeah, I think that's

about good."), while the generated response asked for clarification (*"I don't understand what you mean by '0.5'"*).

- **Conversational Fluidity and Naturalness:** Human responses sometimes involved quick, natural affirmations or suggestions that felt more fluid. For example, in Item ID 77, when the user described a "rough" panning approach, the human expert responded encouragingly with "Yeah, just get real random with it," while the generated response was a simple "Okay, that sounds fine."

- **Actionable Guidance over Generic Analysis:** In cases where the user input was just "Please analyze this audio segment" (Item ID 37, 59), the human expert provided immediate, actionable mixing suggestions ("All right, yeah, bring that way down... solo different things..."), whereas the generated response offered a generic, factual analysis of the audio's musical features ("This audio contains a fast-paced rock piece...").

It is plausible that the structured, detailed, and consistently on-topic nature of the generated responses was perceived as more helpful in certain evaluation contexts, particularly when users sought explicit explanation or guidance. However, the preference is not overwhelming, and the strengths of human interaction—contextual understanding, fluidity, and providing immediately relevant actions—remain evident.

These observations should be considered preliminary due to the sample size (N=100 comparisons across 10 evaluators). Further analysis, potentially involving qualitative coding of responses or larger-scale evaluations, would be needed to draw more definitive conclusions about the factors driving user preference in this co-creative, instructional context.

## F   Real-Time Interaction Study Details

### F.1   Rationale and Methodology

While offline evaluations like LLM-as-a-judge (Appendix D) and pairwise human preference studies (Appendix E) provide valuable insights into the quality of generated responses in isolation, they do not fully capture the user experience of interacting with the AI agent dynamically within a task context. Research in HCI and AI evaluation emphasizes the importance of studying systems in settings that reflect real-world usage to achieve ecological validity (Subramanian et al., 2023; Kumar et al., 2024). Evaluating co-creative systems, in particular, benefits from observing the interaction patterns, user satisfaction, and collaborative dynamics as they unfold (Karimi et al., 2018).

This study aimed to assess user experience and perceived utility when interacting with the co-creative mixing agent in real time. We deployed the fine-tuned Qwen model via a chat interface built on HuggingFace Spaces, which allowed users to upload an audio file and engage in a multi-turn conversation with the agent.

10 music producers (2 expert, 5 intermediate, 3 amateur) participants were recruited. Participants were compensated with a $2.50 Amazon Gift Card for their time. Each producer was provided with a link to the HuggingFace Space. They were instructed to interact with the agent by uploading their audio of choice and using the chat interface to get recommendations and guidance on mixing the track. Participants were asked to continue the interaction iteratively until they felt satisfied with the agent's recommendations or until approximately 5 minutes had passed.

Immediately following the interaction, participants completed a survey designed to capture their subjective experience regarding conversational quality, creative contribution, and overall usability.

### F.2   Survey Questions

The post-interaction survey included the following questions.

- **Q1: What is your level of music production expertise?** (e.g., Amateur, Intermediate, Expert)

- **Q2: How natural did the conversation with the agent feel?** (e.g., on a scale of 1-4, where 1=Very Unnatural, 4=Very Natural)

- **Q3: How much did the agent contribute to the creative process, compared to your own input?** (e.g., on a scale of 1-4, where 1=Significantly less than my own input, 4=Significantly more than my own input)

- **Q4: Did the agent suggest musical ideas that you wouldn't have thought of yourself?** (e.g., on a scale of 1-4, where 1=Not at all, 4=Definitely)

- **Q5: Are there any other insights you noticed during your interaction with the agent?** (Open-ended text response)

### F.3 Quantitative Survey Results

Participants rated their experience on several Likert scales (1-4). The aggregated results (N=10) are summarized below:

| Response | Count | % |
|---|---|---|
| Very natural | 4 | 40.0 |
| Somewhat natural | 3 | 30.0 |
| Somewhat unnatural | 3 | 30.0 |
| Very unnatural | 0 | 0.0 |

(a) Conversation Naturalness (Q2)

| Response | Count | % |
|---|---|---|
| Definitely | 0 | 0.0 |
| Somewhat | 5 | 50.0 |
| A little | 2 | 20.0 |
| Not at all | 3 | 30.0 |

(b) Novelty of Ideas (Q4)

Table 12: Real-Time Interaction Survey Results (N=10): Conversation Naturalness and Novelty of Ideas.

| Response (Agent vs. Own Input) | Count | Percentage (%) |
|---|---|---|
| Significantly more | 0 | 0.0% |
| Slightly more | 4 | 40.0% |
| Slightly less | 5 | 50.0% |
| Significantly less | 1 | 10.0% |

Table 13: Real-Time Interaction Survey Results: Agent Contribution (Q9)

Overall, the quantitative feedback suggests users found the agent reasonably conversational and capable of contributing some novel ideas, but perceived its creative contribution as generally less than their own input.

### F.4 Quantitative Analysis of Audio Understanding Guidance

The qualitative feedback from the real-time study highlighted limitations in the model's ability to provide guidance based on audio context. To explore this further and to better quantify the impact of fine-tuning on MIXASSIST, we conducted a manual analysis of generated responses. One of the authors developed a coding scheme and labeled 100 generated conversations from our fine-tuned Qwen model and 100 from the base Qwen model (without fine-tuning).

This analysis revealed that fine-tuning on MIXASSIST significantly enhances the model's ability to provide substantive, task-relevant guidance. Our fine-tuned model identified a specific audio issue in 36% of responses and proposed a concrete, actionable solution in 41% of responses. In contrast, the base model identified a specific audio issue in only 26% of responses and proposed an actionable solution in just 22% of responses.

Additionally, a targeted analysis was conducted to evaluate the *correctness* of the guidance—that is, whether the model's suggestion aligned with expert expectations for the given context. This evaluation found that the fine-tuned Qwen model proposed correct actionable guidance in 35% of instances. The base Qwen model, without fine-tuning, proposed correct actionable guidance in only 14% of responses.

This direct comparison confirms that while the broader challenge of deep audio understanding and a model's ability to truly act as a teacher remains unsolved (Xie et al., 2025; Gong et al., 2023; Ma et al., 2025), fine-tuning on MIXASSIST demonstrably improves a model's capacity to engage with the music mixing task in a meaningful, relevant, and correct way.

### F.5 Qualitative Feedback Analysis (Q11)

The open-ended question, *"Are there any other insights you noticed during your interaction with the agent?"*, elicited responses from the 10 participants that were thematically analyzed. The following key themes emerged.

#### F.5.1 Audio Analysis Capability (Mixed/Limited)

Throughout this section, we use Participant IDs (e.g., Participant1 (P1)) to refer to the producers who are quoted. The number following the letter refers to the group in which the completed the experiment. A core theme involved the agent's handling of audio. Several participants highlighted limitations or inconsistencies. P5 explicitly noted the agent's inability to gain insight from the track:

> *"It would have been more helpful had it been able to analyze the audio. It said that it didn't have any insight into the track itself."* - P5 (Intermediate)

P1 described varying levels of analysis depth, sometimes feeling superficial:

> *"...the agent wasn't always super clear about how it was analyzing the audio... sometimes it felt more like the agent was reading a description of the audio off a wiki page..."* - P1 (Intermediate)

Specific failures included misidentifying chord progressions (P6) or struggling with tempo detection (P7). Conversely, P9 experienced a moment of insight:

> *"I noticed the agent quickly identified my tendency to over boost the bass and gently suggested a more balanced EQ curve."* - P9 (Expert)

#### F.5.2 Conversational Interaction (Mostly Positive)

The conversational aspect was generally well-received. P6 found it "very natural", P2 found it a "pleasure to use", and P1 appreciated its ability to adapt to non-technical language:

> *"I especially liked how when I used non-specific, non-technical terms to describe a technical problem, the agent matched me by using those same terms in its responses."* - P1 (Intermediate)

However, some negative experiences occurred, such as P5 feeling responses were "sort of canned", P4 noting occasional "confusion", and P7 encountering irrelevant suggestions:

> *"Strange responses as the conversation went on. Such as telling me to go for a walk."* - P7 (Amateur)

#### F.5.3 Quality and Utility of Suggestions

The advice provided was often considered useful, particularly for less experienced users (P6) or as a starting point (P3). The agent demonstrated adaptability in some cases:

> *"It adapted to my preference for a punchy drum sound, offering tailored compression settings."* - P10 (Amateur)

Yet, there was a desire for greater depth or creativity. P5 wanted "more precise or little more out-of-the-box" ideas, and P3 expected different technical advice. A potential technical inaccuracy was also noted by P2 regarding reverb parameters.

### F.5.4 Minor Quirks/Bugs

Small technical glitches were observed, such as P2 finding a "stray non-English character" and P4 needing to repeat questions when the agent got confused.

### F.5.5 Desire for Future Capabilities

Participants suggested some possible future directions. P1 wished for assistance in learning terminology:

> *"I would love if in the future, it could also help me identify technical terms for problems I can't quite identify by name."* - P1 (Intermediate)

P8 expressed interest in generative music capabilities:

> *"It would've been dope to see how it generates beats and suggests melodies."* - P8 (Amateur)

The qualitative feedback from the 10 participants indicates that while the agent shows promise in conversational interaction and providing baseline guidance, significant improvements are needed in its audio analysis capabilities and the depth/creativity of its suggestions.

## G   MIXASSIST Dataset Details

This section provides a detailed description of the MIXASSIST dataset, including its construction, processing, statistical properties, and analysis. MIXASSIST is designed to facilitate research into conversational, co-creative AI for music mixing instruction by capturing audio-grounded, multi-turn dialogue between expert and amateur music producers.

### G.1   Dataset Construction

The foundation of MIXASSIST lies in seven hour-long co-creative mixing sessions conducted with 12 music producers (7 experts, 7 amateurs, forming 7 distinct pairs). Participants were recruited following an initial survey gauging interest in AI mixing assistance (see Appendix B). Participants were compensated for their time in the co-creative mixing sessions, with experts receiving a $36 Amazon Gift Card and amateurs receiving an $18 Amazon Gift Card. This study was IRB approved. During the sessions, amateurs used their preferred Digital Audio Workstation (DAW) to mix multitrack audio recordings provided for the study.

These multitracks were sourced from **The Mix Evaluation Dataset** (De Man & Reiss, 2017), a publicly available collection designed specifically for research in music mixing perception and practices (De Man & Reiss, 2017). This dataset was chosen because it aggregates multitrack sessions from various established sources, including professional recordings released under Creative Commons licenses (allowing reuse for research) and tracks provided by educational resources like Weathervane Music's "Shaking Through" series and Mike Senior's "Mixing Secrets" Multitrack Library. This compilation offers a valuable mix of genres (e.g., rock, pop, funk, country, soul) and recording contexts, providing ecologically valid material essential for the mixing task within our study. Table 15 details the specific song, artist, genre, and expert/amateur participant pair for each of the seven sessions conducted.

| Session ID (Genre) | # Relevant Expert Turns |
|---|---|
| 1 (ROCK) | 50 |
| 2 (POP ROCK) | 93 |
| 3 (FUNK) | 55 |
| 4 (COUNTRY) | 57 |
| 5 (HARD ROCK) | 29 |
| 6 (SOUL) | 45 |
| 7 (INDIE ROCK) | 102 |

Table 14: Number of Relevant Expert Turns per Mixing Session

*Note: Genres correspond to Table 15. Session IDs correspond to the expert/amateur pair number (e.g., Session ID 1 involves E1/A1).*

| Song Title | Artist | Genre | Expert (E) | Amateur (A) |
|---|---|---|---|---|
| Good Time | Louis Cressy Band | ROCK | E1 | A1 |
| I'd Like to Know | Filthybird | POP ROCK | E2 | A2 |
| In the Meantime | Fredy V | FUNK | E3 | A3 |
| Lead Me | The Done Fors | COUNTRY | E4 | A4 |
| Lolita | Purling Hiss | HARD ROCK | E5 | A5 |
| Not Alone | Fredy V | SOUL | E6 | A6 |
| Old Tree | Creepoid | INDIE ROCK | E7 | A7 |

Table 15: Details of the Seven Co-Creative Mixing Sessions

*Note: Songs and artist information sourced from The Mix Evaluation Dataset (De Man & Reiss, 2017). Participant IDs (e.g., E1, A1) correspond to the expert and amateur producers in each session.*

During the sessions, amateurs employed a think-aloud (Van Someren et al., 1994) protocol while receiving guidance from the expert upon request. This setup aimed to capture realistic pedagogical interactions grounded in the audio context. Direct parameter logging was avoided to maintain ecological validity and allow participants their preferred DAW; nuances related to parameters are captured within the dialogue itself.

The approximately 7 hours of session recordings underwent several processing steps to create the final dataset:

- **Transcription:** Initial dialogue transcription was performed using Whisper (Radford et al., 2023).

- **Manual Cleaning & Correction:** One author manually reviewed and corrected the transcripts. Filler words ("uh", "um") were removed, while conversational acknowledgments ("mm-hmm") were retained. Ellipses (...) were used to indicate pauses or sentence fragments characteristic of spontaneous speech. Utterances related solely to experimental timing, researcher questions, or specific DAW interface elements (not mixing practice) were removed.

- **PII Filtering:** During the manual cleaning phase, specific Personally Identifiable Information (PII) mentioned incidentally during the conversations, such as participant names (beyond assigned IDs), specific school or work affiliations, or other unique identifying details, were removed or redacted to protect participant privacy.

- **Speaker Segmentation:** Transcripts were split into amateur and expert utterances.

- **Audio Segmentation:** Relevant music-only audio segments corresponding to the discussion were extracted. This involved capturing the audio played back from the DAW before a conversation turn initiated or the relevant audio context for the dialogue. Care was taken to align audio semantically with the dialogue, though this process has inherent subjectivity. Voices were removed from these segments to ensure only music remained. The average duration of these audio segments is 19.44

seconds indicating the average length of audio playback prior to an amateur asking a question.

- **Placeholder Augmentation:** To handle conversational deviations (e.g., unsolicited expert advice, unanswered amateur questions), standardized placeholders were inserted. "Please analyze this audio segment" was added for the amateur when the expert provided unsolicited feedback on audio. "I need more information before I can respond. Please elaborate." was added for the expert if an amateur's question received no response. This maintained a consistent question-answer structure suitable for model training.

- **Topic Segmentation:** Conversations were segmented into meaningful sub-conversations based on mixing focus (e.g., overall_mix, drums, guitars, keys, vocals). This was achieved using instrument mentions, context, and conversation history, requiring topic switches to span more than one turn and be semantically relevant.

- **Data Filtering for Content:** To ensure the dataset focused on pedagogically useful interactions for training an assistive agent, a filtering stage was applied based on expert response content. A binary metadata tag, *'has_content'*, was initially added to expert utterances to indicate if the response was relevant, addressed the amateur's query, and provided actionable mixing guidance or explanation. This labeling, while inherently subjective, was performed by one author utilizing reflexive practices to maintain consistency (Palaganas & Estacio, 2021; Finlay, 2002). Utterances consisting only of simple affirmations (e.g., "yeah", "cool") or non-instructive filler were deemed detrimental for training a helpful agent and marked as *'has_content'=False*. The conversational pairs where the expert response was marked *'has_content'=True* constitute the final MIXASSIST dataset (N=431), representing 67.34% of the original expert turns. All statistics and analyses, unless otherwise noted, pertain to this filtered dataset.

- **Contextual Augmentation (Topic Boundaries):** To improve continuity between topic segments, `gpt-4o-mini-2024-07-18` was used to generate contextual transitions within the first amateur utterance following a topic change. This synthesized historical context from previous turns while aiming to maintain the amateur's voice, bridging potential gaps created by segmentation. (See Subsection G.3 for the prompt structure). These synthetic augmentations are primarily for model training coherence and are excluded from linguistic analyses reported elsewhere throughout this section.

The complete, unprocessed audio recordings containing both dialogue and DAW playback are also provided to support further research.

## G.2 Experiment Participant Instructions

This section details the guidelines provided to the expert and amateur participants, outlining their roles and the mixing task for the co-creative sessions described in Appendix G.

### G.2.1 Instructions for Amateur Participants

**Task Overview** Your task is to create a complete mix of a song from raw multitrack audio files. You will work alongside a more experienced music producer who acts as a mentor. However, **they will only offer assistance if you directly request it**. While the goal is to achieve a final mix aligned with a specific target genre using your own style, the **primary objective is to learn and improve your mixing skills** through this interaction.

**Materials Provided**

- Raw multitrack audio files (via Zoom).
- A list of allowed audio effects (Gain, Pan, Equalization, Reverb, Compression, Gate, Limiter, Delay, Phaser, Flanger, Chorus). You may use some or all of these.
- Your preferred Digital Audio Workstation (DAW) software (e.g., Reaper, Logic, Ableton) is required.

**Session Procedure**

1. **Import and Setup:** Open your DAW, import the provided raw audio tracks, and note the target genre for the mix (to be specified).

2. **Screen Sharing:** Initiate screen sharing of your DAW window so the researcher and expert can view your workspace.

3. **Mixing Process:** Begin mixing the tracks. You have full creative control over your approach.

4. **Think Aloud Protocol:** While mixing, continuously verbalize your thoughts and decisions. Specifically explain:
   - *Why* you are making adjustments (e.g., "The snare feels lost, so I'm increasing its gain.").
   - *Which* parameters you are changing (e.g., "Adjusting EQ on the bass to boost lows.").
   - *Why* you chose specific values (e.g., "Setting compressor attack to 5ms to let transients through.").

5. **Requesting Assistance:** If you encounter challenges or need guidance, **we strongly encourage you to actively ask the experienced producer for help, clarification, or their opinion**. Engage in follow-up discussion until you reach understanding.

**Important Considerations**

- The expert acts as an **assistor/mentor** and provides help **only upon your request**.

- Experimentation with different mixing techniques is encouraged.

- Focus on the primary goal: **learning and improving mixing skills** by leveraging the expert's guidance.

- The session duration is approximately 75 minutes.

*G.2.2 Instructions for Expert Participants*

**Role Overview**  Your role is to **guide and support** a less experienced music producer during their mixing task. Act as a **mentor**, offering expertise and insights while empowering them to lead the process and make decisions.

**Initial Setup Checks**

- **View Screen Share:** Ensure you can clearly see the amateur participant's shared DAW screen.

- **Confirm Raw Tracks:** Verify the DAW session contains only the raw audio files, without prior edits, to ensure a fresh start.

- **Note Target Genre:** Be aware of the specific target genre for the mix.

**Mentoring Guidelines**

**Encouragement:**  Offer help and support as the amateur sets up their DAW session.

**Active Listening:**  Pay close attention to the amateur's actions within the DAW and their verbalized thought process (think-aloud).

**Responding to Questions:**  Provide detailed, actionable advice when asked. Be specific about techniques, effects, and parameter settings/values where applicable.

- *For Specific Questions:*
    *Example:* Instead of just "Try a compressor," suggest: "Consider adding a compressor on the snare with a 4:1 ratio and 20ms attack. This helps control dynamics for consistency." Aim for specificity.

- *For General Uncertainty:* Offer guiding suggestions but encourage the amateur's choice.

  *Example:* Instead of "Start with drums," suggest: "Often, balancing drums and bass first builds a solid foundation, but feel free to explore other starting points. I'm here to help when you need guidance."

**Explaining Rationale: Always explain the reasoning behind your recommendations** to facilitate learning.

  *Example:* Instead of "Use reverb on pop vocals," explain: "In pop, reverb on vocals adds space. A small room reverb with short decay can achieve this without cluttering the mix."

**Positive Reinforcement:** Maintain a supportive and encouraging tone. Acknowledge successes and frame mistakes as learning opportunities.

**Important Considerations**

- The primary goal is to facilitate the amateur's **learning and growth**, not necessarily to achieve a flawless mix.
- Approach the interaction with patience, understanding, and flexibility.
- Foster a **collaborative and positive learning environment**.
- The session duration is approximately 75 minutes.

### G.3 Context Generation Prompt Structure

To generate the introductory context sentence(s) for amateur utterances at topic boundaries (as described in the Contextual Augmentation step above), the following Python f-string based prompt structure was used with GPT-4o. The variables ("current_topic", "all_previous_turns", "current_first_turn") were populated dynamically for each transition point.

```
prompt = f"""You are the amateur music producer continuing a
    ↪ conversation.
      The current topic is {current_topic}.

      Previous conversations content (most recent 10 turns):
      {json.dumps(all_previous_turns[-10:], indent=2)}

      Your next message will be: "{current_first_turn}"

      Before that message, briefly summarize what was previously
          ↪ discussed that led to this point.
      Write 1-2 sentences in a casual, first-person style as if you're
          ↪ the amateur producer recapping
      what you were just talking about. Focus on technical details and
          ↪ decisions that were made.
      The summary should flow naturally into your next message.
      """
```

Listing 2: Python f-string prompt structure for context generation.

### G.4 Dataset Statistics and Splits

The resulting MIXASSIST dataset comprises 431 substantive, audio-grounded expert responses derived from 7 original mixing sessions. Expert utterances average 36.57 tokens, while the preceding amateur utterances average 25.39 tokens (based on all original amateur turns). The dataset was partitioned into training, development, and test sets using the topic-based sub-conversations (of which there are 77 containing substantive content) as units, rather than splitting individual turns, to preserve conversational integrity and evaluate generalization across interaction dynamics. The splitting strategy aimed to rigorously

evaluate model generalization across different interaction styles and musical contexts (see Table 14 for session turn counts). Specifically, the complete sub-conversations from the shortest (Session ID 5, Hard Rock) and the second longest (Session ID 2, Pop Rock) overall sessions were allocated exclusively to the test set. This preserves the longest session (ID 7, Indie Rock)—allowing its potentially extended dialogues and varied sub-conversations to be represented across all splits—while holding out the distinct genres of Pop Rock and Hard Rock provides diverse test cases against the remaining pool (Funk, Country, Soul, Rock, Indie Rock). Sub-conversations from the remaining five session groups were distributed across the training, development, and test sets. This approach ensures that the test set contains examples from all seven producer pairs and song contexts, while the training and development sets contain data primarily from five pairs. **Consequently, models are frequently evaluated on data from producer pairs and potentially genres unseen during fine-tuning, providing a more robust assessment of generalization capability.** The resulting distribution across the splits is as follows:

- **Train Set:** 241 examples across 28 conversations.
- **Development Set:** 34 examples across 11 conversations.
- **Test Set:** 156 examples across 38 conversations.

This distribution results in varying average turns per conversation across splits (Train: 8.61, Dev: 3.09, Test: 4.11), reflecting the natural variability in topic discussion length and the inclusion of full short/long conversations in the test set.

### G.5   Dataset Instance Examples

This section provides illustrative examples of the data instances contained within the MIXASSIST dataset. Each instance includes context (conversation ID, topic, turn ID), the input audio segment, the preceding conversational history (including a generated summary and recent turns), the current user utterance, and the target expert assistant utterance.

*Example 1: Discussing Keys*

**Conversation ID:** `group4_conv10`

**Topic:**          `keys`

**Turn ID:**          1

**Input Audio File:** `audio/segments/Group4/cut_37.wav`

**Input History:** • *(Summary):* We were discussing how to approach mixing a track with quiet vocals and many guitar tracks. The expert suggested starting with vocals and drums, and then using a gentle glue compressor on the drum bus to add presence and saturation. I think I'll keep balancing out this.
• *(Assistant):* Would you solo the keys and the accordion together?

**User Utterance:** Yeah! The keys are so quiet.

**Assistant (Target) Utterance:** I think I would put a compressor bus on both of those. And turn that limiter on on the compressor. Maybe turn the threshold down a little bit.

**Example 2: Checking Overall Mix Level**

**Conversation ID:** `group6_conv4`

**Topic:**          `overall_mix`

**Turn ID:**          1

**Input Audio File:** `audio/segments/Group6/cut_19.wav`

**Input History:** • *(Summary):* We've been working on adjusting the EQ and compression to give the guitar more space in the mix, making some tweaks in the frequency ranges and considering adding reverb and delay for more

character. Now, I'm about to listen to an audio segment to see how these changes sound. Please analyze this audio segment.

- *(Assistant):* Can we bump the volume a little bit? Just, we don't want it to be too far into red, but even just in terms of like, I feel like, I still can't hear it over the, the others, so I think we have to pull down some of these percussion.

**User Utterance:** Little peaking here.

**Assistant (Target) Utterance:** Yeah, I think it's okay. It's not clipping.

### G.6 Conversational and Linguistic Analysis

**Topic Flow:** The original mixing sessions captured show an average of 10.0 topic switches per session. A common pattern observed is a "zoom in, zoom out" approach, where participants focus on individual elements (like drums) before returning to assess the overall mix, mirroring professional workflows (Emil, 2024). Within the 431 substantive expert turns comprising MIXASSIST, Drums are the most frequent topic (40.4%), followed by overall mix (25.3%), guitars (15.1%), vocals (8.6%), bass (6.5%), and keys (4.2%).

**Linguistic Differences (Expert vs. Amateur):** Analysis reveals nuanced linguistic patterns. Technical term usage, as a percentage of tokens, is remarkably similar between experts (in their content turns) and amateurs (across all their turns) (Expert Content: 5.93%, Amateur All: 6.00%). Experts exhibit a broader unique vocabulary (Shared Vocabulary Size: 660 lemmas, Expert-only: 504, Amateur-only: 360) and produce text with moderately higher complexity (Avg FKGL: Expert Content = 4.40, Amateur All = 2.80). Additionally, experts utilize metaphorical language more frequently than amateurs (Avg Expert Metaphors per content turn: 0.715 vs. Avg Amateur per turn: 0.499, approx. 1.43x rate).

**Instructional Patterns:** Conversations exhibit clear question-explanation structures, with amateurs often asking questions and experts providing explanations. Distinct step-by-step instructional sequences from experts, followed by amateur implementation attempts and feedback cycles, form recognizable pedagogical patterns. %(This paragraph remains valid)

**Temporal Dynamics:** Analysis over session time reveals strong evidence of active learning. Experts introduce technical terms at an average rate of 2.17 per relevant turn. Crucially, amateurs demonstrate significant positive growth in their technical terminology usage, increasing by an average of +42.31% from the first to the last third of their interactions. This indicates active vocabulary acquisition occurring alongside practical skill development during the collaborative mixing process.

### G.7 Availability

The MIXASSIST dataset, including the processed conversational data, associated audio segments, and the raw session recordings, is publicly available to facilitate further research.

## H MIXPARAMS Dataset Details

This appendix details the MIXPARAMS dataset, a resource created as an extension to The Mix Evaluation Dataset (De Man & Reiss, 2017) and serving as a complementary dataset to the primary MIXASSIST conversational corpus presented in this work. While MIXASSIST focuses on the audio-grounded instructional dialogue (the "why"), MIXPARAMS provides detailed technical parameters (the "how") derived from corresponding source material.

### H.1 Motivation and Purpose

MIXPARAMS was created to help advance research in intelligent music production, specifically by providing detailed, ethically sourced, and interpretable data about music mixing parameters within Digital Audio Workstations (DAWs). A primary motivation was to fill a gap created by the scarcity of publicly available, detailed mix session data, as such

data is often proprietary. The dataset facilitates the exploration of tasks such as predicting context-appropriate audio effect parameters (e.g., EQ, compression, reverb settings), potentially linking these technical settings to the high-level perceptual evaluations available in the original Mix Evaluation Dataset or the instructional dialogue captured in MIXAS-SIST. Furthermore, the creation of this structured, tabular dataset enables investigation into whether interpretable machine learning models (e.g., regression, multi-label classification) can achieve strong performance on parameter prediction tasks, offering an alternative to less transparent deep learning approaches like DDSP or graph neural networks often necessitated by lack of such data.

## H.2 Dataset Content and Format

MIXPARAMS consists of detailed annotations for **114 individual mixes**. These mixes were derived from a selection of the non-copyrighted DAW session files originally collected for The Mix Evaluation Dataset. The specific songs included in MIXPARAMS are listed in Table 16. Crucially, the dataset uses the same source multitrack audio stems as The Mix Evaluation Dataset (and thus the same source songs as used in the MIXASSIST collection sessions), primarily sourced from The Open Multitrack Testbed, Weathervane Music's "Shaking Through", and Mike Senior's "Mixing Secrets" library.

| Artist | Song Title |
|---|---|
| The Done Fors | Lead Me |
| Fredy V | In The Meantime |
| Fredy V | Not Alone |
| Broken Crank | Red To Blue |
| The Done Fors | Under A Covered Sky |
| The Done Fors | Pouring Room |
| Filthybird | I'd Like To Know |
| The Districts | Vermont |
| Creepoid | Old Tree |
| Purling Hiss | Lolita |
| Louis Cressy Band | Good Time |

Table 16: Songs from The Mix Evaluation Dataset Included in MIXPARAMS

*Note: These songs were selected from De Man et al. (De Man & Reiss, 2017) based on their non-copyrighted status, allowing for parameter annotation and distribution in* MIXPARAMS.

Each instance in the dataset corresponds to a single track within one of the 114 annotated mixes. The annotations for each track include:

**Metadata:** Basic information including `mix_name`, `song_name`, `artist_name`, and `genre`.

**Track Information:** Details about the specific track within the mix, such as `track_name`, `track_type` (e.g., audio, aux), `track_instrument_type`, `track_instrument_subtype`, and `channel_mode` (mono/stereo).

**Audio File Reference:** Information linking to the source audio, including `track_audio_path` (path to the original stem file), `track_audio_sample_rate`, and measured `track_audio_lufs` (loudness).

**Parameter Data:** A structured representation (field named `parameters`) detailing the audio effects applied to the track and their settings. This includes common effects like Gain, Pan, EQ, Compression, and Reverb, along with their specific parameters (e.g., EQ frequency, gain, Q; Compressor threshold, ratio, attack, release; Pan position).

The dataset focuses on common mixing parameters from native plugin; non-native plugins encountered in the original sessions were omitted during annotation. While the dataset provides parameter settings, it relies on the external availability of the original audio from their respective sources (Open Multitrack Testbed, Weathervane Music, Mixing Secrets).

### H.3 Data Collection and Preprocessing

The parameter data was collected manually by an author of this work. The process involved opening each original DAW session file (Logic Pro or Pro Tools) corresponding to the selected mixes (Table 16) provided by The Mix Evaluation Dataset and meticulously annotating the settings for audio effects applied to each track.

A significant challenge was that some plugin interfaces visually obscure their precise numerical parameter values. To address this, a custom software tool was developed and used to estimate these parameter values based on their visual representation within the plugin GUI. This tool is publicly available. If a parameter's value could not be reliably determined or estimated, it was omitted from the dataset.

**Handling of Automation:** It is important to note that while automation (time-varying parameter changes) is a crucial aspect of mixing, the dynamic automation data itself was not captured in MIXPARAMS due to the complexity of annotation. Instead, if a parameter was automated, the value recorded in the dataset typically represents the parameter's state at the end point of the automation relevant to the mix section or snapshot being annotated. For instance, if a track's volume fader was automated from -4dB to -2dB over a section, the value annotated for that track's gain/volume would be -2dB. This captures the final intended level but not the dynamic transition.

The raw DAW session files themselves contain the full automation data but are not part of the distributed MIXPARAMS dataset; researchers needing the original session files should refer to The Mix Evaluation Dataset. No other significant preprocessing was applied beyond the described annotation and estimation process.

### H.4 Usage Notes

No canonical data splits (train/dev/test) are mandated for MIXPARAMS, and users should define splits appropriate for their specific task. However, for reproducibility or comparison purposes, the dataset hosting includes metadata allowing for partitioning. **One possible partitioning, based on the internal structure where each row represents an annotated track, yields 1.55k rows for training, 300 rows for development, and 500 rows for testing.**

It is noted that not all mixes included in The Mix Evaluation Dataset (and thus potentially annotated in MIXPARAMS) have corresponding perceptual evaluation data (preference ratings or comments). Mixes originating from Mike Senior's "Mixing Secrets" collection, for example, lack these perceptual ratings. Researchers intending to link parameter settings with perceptual data should filter MIXPARAMS accordingly.

### H.5 Ethical Considerations

MIXPARAMS was created with the intention of providing an ethically sourced repository for music mixing research. It derives from publicly available data and session files originally collected by De Man et al. (De Man & Reiss, 2017). As this work focuses on annotating the technical artifacts (parameter settings) from those sessions rather than collecting new data directly from individuals, the original mix engineers were not re-contacted, and an additional IRB review was unnecessary for this specific annotation effort. The dataset contains no direct PII, sensitive, confidential, or offensive information.

