# OpenReview forum: "MixAssist: An Audio-Language Dataset for Co-Creative AI Assistance in Music Mixing"
_colmweb.org/COLM/2025/Conference — COLM 2025_

### Official Review · Reviewer_jqzw · 2025-05-12

**Rating:** 5
**Confidence:** 4
**Ethics Flag:** 1

**Summary:**

The paper "MixAssist: An Audio-Language Dataset for Co-Creative AI Assistance in Music Mixing" introduces MIXASSIST, a novel dataset designed to support the development of AI systems that can provide co-creative assistance in music mixing. The dataset captures the dynamic, multi-turn dialogues between expert and amateur music producers during live mixing sessions, focusing on the instructional and conversational aspects of the mixing process. The authors conducted seven in-depth mixing sessions involving 12 producers, resulting in 431 audio-grounded conversational turns. The dataset is unique in its focus on capturing the "why" behind mixing decisions, providing a rich resource for training AI models to understand and participate in real-world music production dialogues. The paper also evaluates several state-of-the-art audio-language models (ALMs) fine-tuned on MIXASSIST, demonstrating promising results in generating contextually relevant mixing advice.

**Questions To Authors:**

How did the authors ensure the diversity and representativeness of the musical genres and producer backgrounds in the MIXASSIST dataset?

**Reasons To Accept:**

The introduction of the MIXASSIST dataset represents a significant contribution to the field of AI in music production. The dataset is meticulously constructed, capturing the nuances of real-world music mixing dialogues, which makes it a valuable resource for training and evaluating AI models. The focus on co-creative and instructional interactions aligns with the growing interest in AI systems that can provide educational and collaborative support, rather than just automation. The paper provides a comprehensive evaluation of several ALMs, demonstrating the potential of fine-tuning models like Qwen-Audio on MIXASSIST to generate high-quality, contextually relevant mixing advice. The findings highlight the strengths and limitations of current ALMs in this domain, providing valuable insights for future research. The detailed methodology, including participant instructions, data processing steps, and evaluation metrics, ensures the reproducibility and reliability of the study. The release of the dataset, along with the processed data and raw recordings, encourages further research and development in this emerging area.

**Reasons To Reject:**

One potential concern is the limited diversity of the dataset in terms of musical genres and producer backgrounds. While the dataset includes a variety of genres, the overrepresentation of certain genres (e.g., rock, pop) and the use of a specific set of multitrack recordings from The Mix Evaluation Dataset could limit the generalizability of the findings. The authors could benefit from discussing strategies to expand the dataset to include a broader range of musical styles and production contexts. Another issue is the reliance on a single model (Qwen-Audio) as the primary baseline, which may not fully capture the capabilities and limitations of different ALMs in this domain. Including a more diverse set of models in the evaluation could provide a more comprehensive understanding of the challenges and opportunities in developing AI assistants for music mixing. The paper could also benefit from a more detailed discussion on the ethical implications of using AI in creative domains, particularly regarding data provenance, attribution, and the potential impact on human creativity and originality.

---

> ### Author Response · Authors · 2025-06-02
> **Evaluation Methodology, Clarity, and Dataset Diversity**
>
> ### Evaluation Methodology and Clarity
> > @jqzw: "[A weakness is the] reliance on a single model (Qwen-Audio) as the primary baseline..."
>
> Thank you for this opportunity to clarify. Our evaluation was not limited to a single model; for our LLM-as-a-judge evaluation, we benchmarked **three models: Qwen-Audio, LTU, and MU-LLaMA**, chosen for their varied strengths in general audio understanding, audio reasoning, and music-specific processing. As reported in Table 2, Qwen-Audio was the top performer in these rankings and was therefore selected for the more resource-intensive in-situ human study. A key challenge with human experiments in this domain is recruiting participants with very specific qualifications, which makes running in-situ experiments with an arbitrary number of models infeasible. Our finding that even this SOTA model faced significant challenges underscores the difficulty of the task we are investigating.
>
> ### Dataset Diversity
> You raise a valid point about generalizability. The limitations of the source material reflect the general dearth of diverse, high-quality, open-source multitracks for mixing research. By establishing a methodology for capturing situated, pedagogical dialogue, MixAssist serves as a first step and a blueprint for future work to create larger and more diverse datasets in this domain, which is a primary goal of this research.

---

> ### Author Response · Authors · 2025-06-02
> **Ethical Considerations**
>
> > @jqzw: "The paper could also benefit from a more detailed discussion on the ethical implications..."
>
> We agree and thank jqzw for emphasizing this aspect. Ethical considerations were central to MixAssist's design from its inception. As detailed in our paper, our work was grounded in an initial user study to understand producer needs and concerns (Section 3.1, Appendix B), intentionally focused on creating a pedagogical system designed to augment, not replace, human creativity (Abstract; Section 1; Section 4.4), and utilized permissively licensed source material for the mixing sessions (Section 3.2; Appendix G.1). To strengthen further and make our position more explicit in light of your feedback, we are developing a formal, expanded Ethics Statement for inclusion in the final paper. The current iteration of this statement, which addresses the broader societal context informed by recent research, is presented below.
>
> >**Ethics Statement** The development and deployment of MixAssist were guided by ethical considerations central to co-creative AI. All 12 participants provided informed consent for the recording and use of their dialogue and actions for research purposes, and all Personally Identifiable Information (PII) was redacted during processing (Section G.1). The dataset is built upon publicly available multitrack recordings from "The Mix Evaluation Dataset", ensuring no copyright infringement (Section 3.2).
> However, we acknowledge that the development of advanced AI assistants carries broader societal risks that warrant careful consideration. While our pedagogical approach aims to augment human skill, we recognize that such tools could be viewed as a potential replacement for music teachers and a threat to producers who earn income by coaching others. Furthermore, we must be cognizant of the risks to creative originality. Research in the parallel domain of creative writing by Anderson et al. (2024) has shown that using LLMs can lead to users producing "less semantically distinct ideas...at the group level". This highlights the challenge, as Chakrabarty et al. (2025) articulate, that "AI writing assistants must enhance human creativity and expression rather than homogenize content or diminish linguistic diversity". This ties into the concern of artists potentially losing their unique voice if AI tools overshadow personal style—a key issue their work addresses by aiming to "mitigat[e] Idiosyncrasies and Improv[e] Human-AI Alignment".
> Despite these valid concerns, we believe the potential benefits of well-designed, human-centric AI assistants are significant. The goal of MixAssist is not to prescribe a single correct path, but to help in the creation of agents that act as collaborative partners that can scaffold learning, demystify complex technical concepts, and empower artists to overcome creative blocks. This aligns with the discussion by Gabriel et al. (2024) in "The Ethics of Advanced AI Assistants," which details numerous opportunities for AI to function as "creative partners, research assistants, counsellors, [and] companions", provided that robust safeguards and ethical alignment are prioritized. Additionally, the work by Chakrabarty et al. (2025) on "salvaging" AI writing through expert-informed edits and improved alignment shows a pathway to refining these tools. By focusing on collaborative, human-in-the-loop models, as MixAssist does, our aim is to develop systems that empower users to develop their unique artistic voice with greater confidence and skill, fostering a co-creative process that values both human ingenuity and AI support.
>
> _Barrett R Anderson, Jash Hemant Shah, and Max Kreminski. 2024. Homogenization Effects of Large Language Models on Human Creative Ideation. In Proceedings of the 16th Conference on Creativity & Cognition (C&C '24). Association for Computing Machinery, New York, NY, USA, 413–425. https://doi.org/10.1145/3635636.3656204_
>
> _Chakrabarty T, Laban P, Wu CS. Can AI writing be salvaged? Mitigating Idiosyncrasies and Improving Human-AI Alignment in the Writing Process through Edits. arXiv preprint arXiv:2409.14509v5. March 4, 2025._
>
> _Gabriel, I., Manzini, A., Keeling, G., Hendricks, L. A., Rieser, V., Iqbal, H., ... & Manyika, J. (2024). The Ethics of Advanced AI Assistants. Google DeepMind; 2024. arXiv preprint arXiv:2404.16244._

---

### Official Review · Reviewer_UNoT · 2025-05-12

**Rating:** 8
**Confidence:** 5
**Ethics Flag:** 1

**Summary:**

The paper presents a dataset of collaborative mixing sessions involving dialog between an expert and amateur mixing engineer. The dataset is used to finetune and evaluate LLMs on their ability to act as such an expert instructor, primarily for pedagogical reasons.

**Questions To Authors:**

- For commercialization, I imagine the most natural setting is a customer (instead of an amateur engineer) interacting with an expert (which could be an AI). The customer may also say things like (as Fig1) "I want the drums to blend better" but also vague things like "It sounds so empty." How do you think the customer's utterances (e.g., request to revise when hiring a mixing engineer) would differ from amateurs'? What is the outlook of applying it in a customer-facing application?
- While prompting is not experimented with in 4.1, I think there is still value in trying that just to understand how bad the models are and what the gaps are. Finetuning inevitably overfits to the bias of the dataset, and decreases outlook for generalization.

**Reasons To Accept:**

- The paper takes a very innotative approach to both the task and methodology of AI-aided mixing. Notwithstanding the motivation of training amateur engineers via dialogs, the data collection methodology of recording reqests and actions in the form of "think aloud" dialogs circumvent the known challenge in collecting audio engineering data.
- One potential that the paper does not emphasize enough is the "what's next" after the expert explains about a concrete action after the amateur describes a desired change. In reality, this can not only help amateurs learn but also directly enable users to mix, even without the model interfacing with the DAW. For example, if an LLM can generate an expert-utterance such as "I would cut 500Hz", the user can just do this and see if it works. For most producers with little experience in mixing, figuring out "I would cut 500Hz" is the hardest part.
- The survey of real practitioners in the industry described in 3.1 is very appreciated.
- The data collection protocol in 3.2 is clearly described.

**Reasons To Reject:**

- The dataset does not come with the submission. I would like to hear the sessions to ensure the validity of the data.
- In Section 4, it is unclear what the input (which utterances are included as the conversation history?) and output of the model (does the model outputs an utterance? even when the atterance is not actionable?) is.
- While the qualitative analysis in 4.3 and 4.4 is helpful, qualitative analyis is lacking. For example, "has conten tag indicating substantive, actionable guidance" previously described can enable quantitative evaluation of models (e.g., did it correct predict that it should EQ the bass? did it correctly identify the offending frequency to cut?).
- "Challenge 1: Audio Understanding" deserves much more discussion, as one could doubt if the models actually did any thing useful based on that section. Is AI simply serving as a chatbot here, if it cannot accurately translate requests to parameters? Can AI actually identify issues in a mix given audio (I don't think this has been thoroughly studied)? If AI cannot make most of the mixing decisions correctly, how can it possibly act as a teacher? To me, this is the **most important weakness of the paper**.

---

> ### Author Response · Authors · 2025-06-02
>
> ### Access to the MixAssist Dataset
> >@UNoT: "The dataset does not come with the submission. I would like to hear the sessions to ensure the validity of the data."
>
> We thank you for emphasizing this, and we apologize for not providing access with the initial submission. To resolve this, we have made the full dataset, including all audio sessions and annotations, available to reviewers via a private link for the duration of the review period. As stated in the paper, we are committed to a full public release of MixAssist upon publication.  **The dataset can be found [at this link](https://zenodo.org/records/15571750?token=eyJhbGciOiJIUzUxMiIsImlhdCI6MTc0ODgyODUzMywiZXhwIjoxNzU0MDA2Mzk5fQ.eyJpZCI6IjAxMzFjODllLTk2OGEtNGZiNS05ZWE1LWE2ZmIxMWFmYzkzZSIsImRhdGEiOnt9LCJyYW5kb20iOiIyOWQ3OWE3OTUyYzk4YjM3MGFjMzBmNDRjOWYyMDc2NSJ9.YaP76WrxjH82pmQoSfopmIf1-HqYSO0gJu0zHgtRJwCgXVsPIrBqj-xEjE616tlky3ai6Cnf9Z0ZcVI2Jnusbw)**. Please note that for the sake of brevity, only one of the group audio segments is presented here.  There are over 700 audio files corresponding to the audio paths shown in the train/test/validation csv data.
>
> ### Evaluation Methodology and Clarity
> > @UNoT: "In Section 4, it is unclear what the input (which utterances are included as the conversation history?) and output of the model (does the model outputs an utterance? even when the atterance is not actionable?) is."
>
> We apologize for this lack of clarity and for not making the specifics of the input construction sufficiently clear in the initial draft of our paper. To directly address this, we have revised Section 3.2 of our paper (detailing Dataset Construction & Analysis) to explicitly state how the 'has_content' filter was applied for constructing model inputs. The revised text for Section 3.2, with additions bolded, is as follows:
>
> >"A critical step involved filtering for pedagogical value: Expert responses were manually annotated with a binary ‘has content‘ tag indicating substantive, actionable guidance. **This tag was crucial not only for selecting target expert outputs but also for curating the expert utterances within the conversational history provided as input to the models. Specifically, expert turns within the input dialogue history were also filtered to include only those marked ‘has content=True’; this consistent approach was adopted to steer the future model's learning towards generating similarly actionable and helpful feedback, rather than general statements about the audio or mixing session.** While inherently subjective, this labeling employed reflexive practices (Palaganas & Estacio, 2021; Finlay, 2002) to maintain consistency. The final MIXASSIST instances exclusively use expert responses marked ‘has content=True‘ as the target output, ensuring the dataset focuses on meaningful instructional content."
>
> ### Additional Experiments
> > @UNoT: "While prompting is not experimented with in 4.1, I think there is still value in trying that..."
>
> This is a great point. We ran a new evaluation comparing our fine-tuned Qwen-Audio model against a zero-shot (prompted) version. Our LLM judge found that the fine-tuned model's responses were preferred in 58% of cases, providing evidence that fine-tuning on MixAssist is necessary for this task, as standard prompting is unlikely to capture the specific nuances of pedagogical mixing dialogue since such conversations aren’t available on the Web or a part of typical post-training data.
> | Metric | Judge | Qwen_FT | Qwen_Base |
> |--------|--------|---------|-----------|
> | **Avg. Rank** | o3-mini | **1.42** | 1.58 |
> | **(Lower is better)** | Qwen3:4b | **1.38** | 1.62 |
> |  | Gemma3 | **1.45** | 1.57 |
> |  | Llama3.1 | **1.47** | 1.53 |
> | **Times Ranked #1** | o3-mini | **144 (57.6%)** | 106 (42.4%) |
> |  | Qwen3:4b | **156 (62.4%)** | 94 (37.6%) |
> |  | Gemma3 | **138 (55.2%)** | 108 (43.2%) |
> |  | Llama3.1 | **133 (53.2%)** | 117 (46.8%) |
>
> ### Commercialization
> This is an excellent question about customer-facing applications. We anticipate that customer language may be more abstract than the amateur-producer dialogue captured in MixAssist. While the dataset's focus on technical and pedagogical dialogue could serve as a foundational resource, adapting to a commercial domain would present new research questions. Future work could explore methods for handling less-technical user intents or augmenting the dataset with dialogues reflecting a customer-client dynamic.

---

> ### Author Response · Authors · 2025-06-02
> **On the Challenge of Audio Understanding**
>
> >@UNoT: "'Challenge 1: Audio Understanding' deserves much more discussion... If AI cannot make most of the mixing decisions correctly, how can it possibly act as a teacher? To me, this is the most important weakness of the paper."
>
> We agree that current ALMs have limitations in this area, which our own user studies confirmed. As we report in Section 4.4, several participants in our real-time study noted that the agent sometimes failed to provide meaningful insights based on the audio, with one stating, _"It would have been more helpful had it been able to analyze the audio"_.
>
> Our goal with this paper is not to claim we have solved this research challenge. Rather, a **key contribution of MixAssist is that it creates an environment to identify, diagnose, and ultimately address this very limitation**. For instance, recent explorations like LLM2Fx Investigate leveraging Large Language Models to predict audio effect parameters directly from textual descriptions without task-specific fine-tuning (Doh et al. 2025). This aligns with trends seen in works such as Chu et al.'s (2025) "Text2FX", which also harnesses embeddings for text-guided audio effect generation. Other approaches, such as Yang et al.’s (2022) "Don't Separate, Learn to Remix", aim for end-to-end neural remixing, seeking to bypass intermediate separation steps for direct remixing. While these are valuable advancements in automation and direct control, they typically do not center on the multi-turn, explanatory dialogue that MixAssist captures for co-creative and pedagogical purposes.
>
> Additionally, the broader challenge of achieving nuanced audio understanding and complex reasoning within general Audio Language Models (ALMs) remains significant. Research by Gong et al. (2023) highlights that even ALMs proficient in general audio tasks may struggle with "fine-grained audio understanding" and "specialized reasoning tasks that require fine-grained understanding," such as temporal reasoning. Similarly, Ma et al. (2025) point to limitations in the reasoning capability of current ALMs for more complex audio queries, suggesting this is partly due to the "simplicity of existing audio datasets," which may not adequately facilitate training for intricate, multi-step reasoning. MixAssist aims to provide richer, context-specific conversational data to help address these gaps, particularly in the domain of co-creative music mixing.
>
> To better quantify our model's behavior and the impact of fine-tuning on MixAssist for this rebuttal, we conducted a manual analysis of the types of generations provided by both our fine-tuned model and the base model (without fine-tuning). One of the authors developed a coding scheme and labeled 100 generated conversations from each model. This analysis revealed that fine-tuning on MixAssist significantly enhances the model's ability to provide substantive, task-relevant guidance.
>
> Our fine-tuned model (on MixAssist) **identified a specific audio issue in 36.00%** of responses and proposed a concrete, **actionable solution in 41.00%** of responses. In contrast, the base model (without fine-tuning on MixAssist) **identified a specific audio issue in only 26.00%** of responses and **proposed an actionable solution in just 22.00%** of responses. This direct comparison confirms that while the broader challenge of deep audio understanding remains unsolved, fine-tuning on MixAssist improves the model's capacity to engage with the music mixing task in a meaningful and relevant way. This makes the dataset a valuable testbed for future improvements in this area of audio understanding.
>
> _Chu A, O'Reilly P, Barnett J, Pardo B. Text2fx: Harnessing clap embeddings for text-guided audio effects. In: ICASSP 2025 - 2025 IEEE International Conference on Acoustics, Speech and Signal Processing (ICASSP). IEEE; 2025:1-5._
>
> _Doh, S. et al. (2025) Can large language models predict audio effects parameters from natural language?, arXiv.org. Available at: https://arxiv.org/abs/2505.20770_
>
> _Gong R, Ma X, Yang Y, Liu Z, Shang L, Jiang M. Enhancing Temporal Understanding in Audio Question Answering for Large Audio Language Models. In: Findings of the Association for Computational Linguistics: NAACL 2024. Association for Computational Linguistics; 2024:405-418._
>
> _Ma X, Yang Y, Gong R, Liu Z, Shang L, Jiang M. Improving Reasoning Capability in Large Audio Language Models. arXiv preprint arXiv:2402.01312. February 2, 2024._
>
> _Haici Yang, Shivani Firodiya, Nicholas J. Bryan, and Minje Kim, “Don’t Separate, Learn to Remix: End-to-End Neural Remixing with Joint Optimization,” in Proceedings of the IEEE International Conference on Acoustics, Speech, and Signal Processing (ICASSP), Singapore, May 22-27, 2022_

---

> > ### Author Response · Authors · 2025-06-02
> > **On the Challenge of Audio Understanding (Cont.)**
> >
> > >@UNoT: "'While the qualitative analysis in 4.3 and 4.4 is helpful, qualitative analyis is lacking. For example, 'has conten tag indicating substantive, actionable guidance' previously described can enable quantitative evaluation of models (e.g., did it correct predict that it should EQ the bass? did it correctly identify the offending frequency to cut?)."
> >
> > Addressing your suggestion for more quantitative evaluation, particularly concerning whether the model provides correct actionable guidance (e.g., "did it correct predict that it should EQ the bass? did it correctly identify the offending frequency to cut?"), we conducted an additional targeted analysis on 100 generated responses from both the fine-tuned Qwen model and the base Qwen model. This evaluation, performed by one of the authors, was used to determine if the generated response identified the substantive issue and followed closely the expected expert guidance regarding target frequencies or parameters.
> >
> > Our fine-tuned Qwen model (on MixAssist) correctly identified the substantive **audio-related issue implied by the amateur's context in 38.00%** of cases where the issue could be addressed and proposed the **correct actionable guidance** (aligning with expert expectations for that context) in **35.00%** of instances The base Qwen model, without fine-tuning on MixAssist, **correctly identified the substantive issue** only **25.00%** of the time and proposed **correct actionable guidance** in **14.00%** of responses.
> >
> > These results provide further quantitative evidence that fine-tuning on the MixAssist dataset not only increases the model's propensity to provide actionable feedback but also significantly improves the correctness and contextual relevance of that guidance. While these figures demonstrate a marked improvement with fine-tuning, they also indicate that there is still a significant way to go in achieving consistently correct and expert-level audio understanding and guidance, a challenge our dataset aims to help the community address. This directly addresses the desire for metrics evaluating the accuracy of the model's substantive suggestions, demonstrating MixAssist's value in training models for such nuanced, expert-level tasks.

---

> ### Comment · Reviewer_UNoT · 2025-06-03
>
> I appreciate the additional qualitative analysis the authors performed. They are informative and clearly worth adding to the paper. I'll maintain my stance of clear acceptance with an increase in the rating.

---

### Official Review · Reviewer_ThtW · 2025-05-14

**Rating:** 9
**Confidence:** 4
**Ethics Flag:** 1

**Summary:**

This work presents MixAssist an audio-language dataset capturing the natural and nuanced multi-turn dialog between an expert and an amateur music producer during collaborative music mixing sessions. This work aims to aid the development of collaborative AI-assisted tools that can help amateur music producers seeking to learn by doing. The dataset consists of 7 hour-long sessions involving 12 producers in total. Specifically, a sample in the dataset contains summary of relevant historical context, the dialogues in the conversation (and sub-conversations), and the associated music-only audio segment. They also performed experiments where they benchmarked finetuned versions of different audio language models (ALMs) (using LoRA) such as Qwen-Audio, LTU, and MU-LLaMA on the proposed MixAssist dataset and evaluated the feedback given by these models using o3-mini as a judge, with Qwen outperforming other models, and human evaluation showed preference of the generated feedback over human annotations, which the authors speculate might be due more detailed explanations (longer response) provided by the model. The paper also includes results for real-time interaction of artists and the system where they majorly found the system to be both natural and helpful.

Overall, this is a very well-written paper where each aspect is carefully thought of and executed, from the process to creating the dataset to taking decisions on how to do it. This paper would without a doubt be very valuable to the community.

**Reasons To Accept:**

- I really appreciate that the authors chose to work on the collaborative and pedagogical aspect of music creation/mixing rather than developing yet another method to automate this creative process.
- This paper presents deep insights about what works with LLMs on this task and what doesn't -- certainly rare these days in papers. An in general, every section is well-motivated and eloquently written.

In my opinion, this paper ticks every check-box on what a good research paper should look like. It was a joy to read this paper and its discussions, so I just want to thank the authors for that.

**Reasons To Reject:**

- If I had to point out something, please add results for a strong open-source model as the judge -- we now have Qwen3 and even Nvidia's nemotron Llama 3.1 235B (which is better than LLaMA 4 Behemoth on some tasks and on par with Deepseek R1).

---

> ### Author Response · Authors · 2025-06-02
> **Additional Experiments**
>
> > @ThtW: "please add results for a strong open-source model as the judge..."
>
> Thank you for this suggestion! We have run a new experiment using gemma3, qwen3, and llama3.1 as the judges to re-evaluate our three baseline models. The new results align with our original findings with Qwen-Audio still ranking first, confirming the use of Qwen-Audio as the model within our human evaluation. These results will be added to the final paper.
>
> | Metric | Judge | Qwen | LTU | MU-LLaMA |
> |--------|--------|------|-----|----------|
> | **Avg. Rank** | o3-mini | **1.59** | 1.7 | 2.71 |
> | **(Lower is better)** | Qwen3:4b | **1.62** | 1.79 | 2.59 |
> |  | Gemma3 | **1.74** | 1.87 | 2.38 |
> |  | Llama3.1 | **1.89** | 1.91 | 2.2 |
> | **Times Ranked #1** | o3-mini | **126 (50.4%)** | 111 (44.4%) | 13 (5.2%) |
> |  | Qwen3:4b | **125 (50.0%)** | 101 (40.4%) | 24 (9.6%) |
> |  | Gemma3 | **113 (45.2%)** | 98 (39.2%) | 38 (15.2%) |
> |  | Llama3.1 | **95 (38.0%)** | 87 (34.8%) | 68 (27.2%) |

---

> > ### Comment · Reviewer_ThtW · 2025-06-07
> >
> > Thank you, glad to see these numbers.

---

### Author Response · Authors · 2025-06-02

We sincerely thank all reviewers for their thorough and highly constructive feedback. We are very encouraged that you found our work to be a **"significant contribution" (jqzw)** that is **"very valuable to the community" (ThtW)**. We are pleased that reviewers highlighted the **"very innovative approach" (UNoT)**, the **"meticulously constructed" (jqzw)** dataset that captures **"deep insights" (ThtW)**, and the focus on the **"collaborative and pedagogical aspect" (ThtW)** of music mixing.

We have carefully considered all comments and have taken immediate action to address reviewers’ primary concerns, including new experiments, new analyses, and providing access to the dataset. Below, we address each of the key points raised by the reviewers, detailing our new findings and clarifications in response to their feedback.

---

### Decision · Program_Chairs · 2025-07-08

**Decision:**

Accept

**Comment:**

This paper introduced a new audio-language dataset for natural and nuanced multi-turn dialog between an expert and an amateur music producer during collaborative setting. All the three reviewers have agreed on the novelty and valuable contribution of this dataset. While one reviewer (#jqzw) has raised ethic and baseline concern, the authors have carefully provided feedback to address them. I thus recommend Accept as is.